# Global retrieval of TROPOMI tropospheric HCHO and NO₂ columns with improved consistency based on updated Peking University OMI NO₂ algorithm

Yuhang Zhang[1,2], Huan Yu[2], Isabelle De Smedt[2], Jintai Lin[1], Nicolas Theys[2], Michel Van Roozendael[2], Gaia Pinardi[2], Steven Compernolle[2], Ruijing Ni[3], Fangxuan Ren[1], Sijie Wang[1], Lulu Chen[1], Jos Van Geffen[4], Mengyao Liu[4], Alexander M. Cede[5,6], Martin Tiefengraber[6], Alexis Merlaud[2], Martina M. Friedrich[2], Andreas Richter[7], Ankie Piters[4], Vinod Kumar[3,8], Vinayak Sinha[8], Thomas Wagner[3], Yongjoo Choi[9], Hisahiro Takashima[10], Yugo Kanaya[11], Hitoshi Irie[12], Robert Spurr[13], Wenfu Sun[2], Lorenzo Fabris[2]

[1]Laboratory for Climate and Ocean-Atmosphere Studies, Department of Atmospheric and Oceanic Sciences, School of Physics, Peking University, Beijing 100871, China
[2]Royal Belgian Institute for Space Aeronomy, BIRA-IASB, Ringlaan 3, 1180 Uccle, Belgium
[3]Max Planck Institute for Chemistry, Mainz 55128, Germany
[4]Royal Netherlands Meteorological Institute (KNMI), Utrechtseweg 297, 3730 AE De Bilt, the Netherlands
[5]NASA Goddard Space Flight Center, Greenbelt, MD 20771, USA
[6]Luftblick, Innsbruck, Austria
[7]Institute of Environmental Physics (IUP), University of Bremen, Otto-Hahn-Allee 1, Bremen 28359, Germany
[8]Department of Earth and Environmental Sciences, Indian Institute of Science Education and Research, Mohali, Punjab, India
[9]Hankuk University of Foreign Studies, Yongin, Republic of Korea
[10]Fukuoka University, Fukuoka, Japan
[11]Research Institute for Global Change, Japan Agency for Marine-Earth Science and Technology (JAMSTEC), Yokohama, Japan
[12]Center for Environmental Remote Sensing, Chiba University (Chiba U), Chiba, Japan
[13]RT Solutions Inc., Cambridge, MA 02138, USA

*Correspondence to*: Jintai Lin (linjt@pku.edu.cn), Michel Van Roozendael (michel.vanroozendael@aeronomie.be)

**Abstract.** The TROPOspheric Monitoring Instrument (TROPOMI), onboard the Sentinel-5 Precursor (S5P) satellite launched in October 2017, is dedicated to monitoring the atmospheric composition associated with air quality and climate change. This paper presents the global retrieval of TROPOMI tropospheric formaldehyde (HCHO) and nitrogen dioxide (NO₂) vertical columns using an updated version of the Peking University OMI NO₂ (POMINO) algorithm, which focuses on improving the calculation of air mass factors (AMFs). The algorithm features explicit corrections for the surface reflectance anisotropy and aerosol optical effects, and uses daily high-resolution (0.25°×0.25°) a priori HCHO and NO₂ profiles from the Global Earth Observing System Composition Forecast (GEOS-CF) dataset. For cloud correction, a consistent approach is used for both HCHO and NO₂ retrievals, where (1) the cloud fraction is re-calculated at 440 nm using the same ancillary parameters as those used in the NO₂ AMF calculation, and (2) the cloud top pressure is taken from the operational FRESCO-S cloud product.

The comparison between POMINO and reprocessed (RPRO) operational products in April, July, October 2021 and January 2022 exhibits high spatial agreement, but RPRO tropospheric HCHO and NO₂ columns are lower by 10% to 20% over polluted regions. Sensitivity tests with POMINO show that the HCHO retrieval differences are mainly caused by different aerosol correction methods (implicit versus explicit), prior information of vertical profile shapes and background corrections; while the NO₂ retrieval discrepancies result from different aerosol corrections, surface reflectances and a priori vertical profile shapes as well as their non-linear interactions. With

explicit aerosol corrections, the HCHO structural uncertainty due to the cloud correction using different cloud
parameters is within $\pm$ 20%, mainly caused by cloud height differences. Validation against ground-based
measurements from global Multi-Axis Differential Optical Absorption Spectroscopy (MAX-DOAS) observations
and the Pandonia Global Network (PGN) shows that in April, July, October 2021 and January 2022, POMINO
retrievals present a comparable day-to-day correlation but a reduced bias compared to the RPRO products (HCHO:
$R = 0.62$, NMB $= -30.8\%$ versus $R = 0.68$, NMB $= -35.0\%$; $NO_2$: $R = 0.84$, NMB $= -9.5\%$ versus $R = 0.85$,
NMB $= -19.4\%$). An improved agreement of HCHO/$NO_2$ ratio (FNR) with MAX-DOAS and PGN measurements
based on POMINO retrievals is also found (NMB: $-14.8\%$ versus $-21.1\%$). Our POMINO retrieval provides a
useful source of information particularly for studies combining HCHO and $NO_2$.

## 53  1 Introduction

Formaldehyde (HCHO) and nitrogen dioxide ($NO_2$) are important trace gases in the troposphere. They play a
critical role in the processes of tropospheric ozone ($O_3$) and aerosol formation, and have significant influences on
air quality, climate and human health (Beelen et al., 2014; Crutzen, 1970; Shindell et al., 2009). Methods to
retrieve tropospheric HCHO and $NO_2$ vertical column densities (VCDs), respectively in the ultraviolet (UV) and
visible (VIS) spectral ranges, have rapidly developed in the last decades, based on sensors mounted on both sun-
synchronous and geostationary satellites such as the Global Ozone Monitoring Experiment (GOME; Burrows et
al., 1999), SCanning Imaging Absorption spectroMeter for Atmospheric CHartographY (SCIAMACHY;
Bovensmann et al., 1999), Ozone Monitoring Instrument (OMI; Levelt et al., 2006), Global Ozone Monitoring
Experiment-2 (GOME-2; Callies et al., 2000), Ozone Mapping and Profiling Suite Nadir Mapper (OMPS-NM;
Dittman et al., 2002), TROPOspheric Monitoring Instrument (TROPOMI; Veefkind et al., 2012), Environmental
Trace Gases Monitoring Instrument (EMI; Zhang et al., 2020), Geostationary Environment Monitoring
Spectrometer (GEMS; Kim et al., 2020) and Tropospheric Emissions: Monitoring of Pollution (TEMPO;
Zoogman et al., 2017). Such satellite observations have been extensively used in studies related to long-term trend
and variabilities (De Smedt et al., 2010; Jiang et al., 2022; Richter et al., 2005), estimation of surface-level
concentrations (Cooper et al., 2022; Wei et al., 2022), constraining emissions of non-methane volatile organic
compounds (NMVOCs) and nitrogen oxides ($NO_x \equiv NO + NO_2$) (Kong et al., 2022; Lin, 2012; Stavrakou et al.,
2018), non-linear ozone chemistry (Jin et al., 2017, 2023; Jin and Holloway, 2015) and impacts on the environment
and human health (Chen et al., 2022; Li et al., 2023).
The retrieval algorithms of tropospheric HCHO and $NO_2$ VCDs based on observations from spaceborne
instruments share many retrieval concepts. First, the slant column density (SCD) representing the trace gas
concentration integrated along the average light path is obtained by performing a spectral fit from backscattered
radiance and irradiance spectra. Second the SCD is converted to a VCD using air mass factors (AMFs) obtained
from radiative transfer (RT) calculations, which are a function of the observation geometry, cloud information,
aerosol properties, surface conditions and the shape of a priori vertical profiles. The main intrinsic differences
between HCHO and $NO_2$ retrievals are that (1) different wavelength ranges are used for each retrieval, and (2) the
final tropospheric HCHO VCDs are determined with additional background correction based on modelled HCHO
columns in the reference region in the Field of Regard (FOR) of satellite instruments, while for $NO_2$ a stratosphere-
troposphere separation is performed in order to obtain tropospheric columns.
Many studies have focused on improving or developing retrieval algorithms to generate scientific HCHO or $NO_2$
products for comparison with operational products and for applications. For example, Liu et al. (2021) present an
improved tropospheric $NO_2$ retrieval algorithm from TROPOMI measurements over Europe, which employs a
new stratosphere-troposphere separation and updated auxiliary parameters, including a more realistic cloud
treatment, for AMF calculation. Over East Asia, Liu et al. (2020) release a new TROPOMI product for
tropospheric $NO_2$ columns that features explicit aerosol corrections in the AMF calculation, and Su et al. (2020)
improve the TROPOMI tropospheric HCHO retrieval by optimizing the spectral fit and using a priori profiles
from a higher resolution regional chemistry transport model.
However, little attention has been paid to fixing the systematic differences in ancillary parameters between HCHO
and $NO_2$ AMF calculations. For instance, the TROPOMI reprocessed (RPRO) HCHO version 2.4.1 and $NO_2$
version 2.4.0 operational products make use of cloud information from different sources: the Optical Cloud
Recognition Algorithm/Retrieval of Cloud information using Neural Networks (OCRA/ROCINN) - Cloud as
Reflecting Boundaries (CRB) product is used for HCHO, while the Fast Retrieval Scheme for Clouds from
Oxygen absorptions bands - Sentinels (FRESCO-S) product is used for $NO_2$. Besides, the surface albedo used in
the current HCHO retrieval is the OMI-based monthly minimum Lambertian-equivalent reflectivity (MLER) at
340 nm with a spatial resolution of 0.5º × 0.5º (lat. × long.), whereas the one used in the $NO_2$ retrieval has been
updated with the KNMI TROPOMI directionally dependent Lambertian-equivalent reflectivity (DLER) v1.0
database at 440 nm with a spatial resolution of 0.125º × 0.125º. Finally, the radiative transfer model used for
HCHO AMF calculation is the linearized pseudo-spherical scalar and vector discrete ordinate radiative transfer
code (VLIDORT) version 2.6, whereas that used for $NO_2$ AMF calculation is the Double-Adding KNMI (DAK)
polarized radiative transfer code version 3.2. Such inconsistencies are an important limitation for studies
combining satellite HCHO and $NO_2$ products, such as analysis of ozone chemistry and wildfires (Jin et al., 2020,
2023). Therefore, there is a need for consistent retrievals of tropospheric HCHO and $NO_2$ VCDs. Moreover, the
TROPOMI operational HCHO and $NO_2$ products do not explicitly account for the optical effect of aerosols, and
use a priori profile shapes from the massively parallel version of the Tracer Model 5 (TM5-MP; Williams et al.,
2017) with a relatively coarse spatial resolution (1º × 1º).
The Peking University OMI $NO_2$ (POMINO) algorithm offers a potential tool to address these limitations.
Founded by Lin et al. (2014), POMINO has been continuously developed and applied to the OMI, TROPOMI and
GEMS instruments (Lin et al., 2014, 2015; Liu et al., 2019, 2020; Zhang et al., 2023). POMINO features an
explicit treatment of aerosol optical effects and surface reflectance anisotropy, as well as a re-calculation of cloud
information using ancillary parameters consistent with those used for $NO_2$ AMF calculation. A smaller bias of
POMINO $NO_2$ data than the operational products has been reported from validation against independent ground-
based measurements (Liu et al., 2019, 2020; Zhang et al., 2023). However, the previous POMINO-TROPOMI
algorithm was limited to Asia, and its potential for HCHO retrieval remained unexplored.
In this paper, we present the global retrieval of TROPOMI tropospheric HCHO and $NO_2$ VCDs with much
improved consistency, based on an updated version of the POMINO algorithm. After describing the methods and
data in Section 2, we present the quantitative comparison of tropospheric HCHO and $NO_2$ columns between
POMINO and RPRO products (Sect. 3). We then discuss the structural uncertainty of HCHO and $NO_2$ retrieval
based on the POMINO algorithm, by conducting a series of sensitivity tests on cloud correction, aerosol correction,
surface reflectance and a priori profile shapes (Sect. 4). Tentative estimates of POMINO retrieval uncertainty are
given in Sect. 5. Finally, we use independent ground-based measurements from a global network of Multi-Axis
Differential Optical Absorption Spectroscopy (MAX-DOAS) instruments and the Pandonia Global Network
(PGN) to validate the tropospheric HCHO and $NO_2$ columns from the POMINO and RPRO products (Sect. 6).

## 2 Method and data

### 2.1 TROPOMI instrument and operational algorithms for HCHO and $NO_2$ retrieval

TROPOMI is an imaging spectrometer onboard the European Space Agency (ESA) Copernicus Sentinel-5
Precursor (S5P) satellite launched on 13th October 2017, crossing the equator at around 13:30 local time (LT)
(Veefkind et al., 2012). Its wide spectral range includes the ultraviolet (UV), visible (VIS), near-infrared (NIR)
and shortwave infrared (SWIR), allowing monitoring of atmospheric trace gases, aerosols, clouds and surface
properties. The original spatial resolution of about 7 km × 3.5 km (along-track × across-track) at nadir was refined
to about 5.5 km × 3.5 km on the 6th of August 2019 by means of a reduction of the along-track integration time.
The wide swath of about 2600 km in the across-track direction enables global coverage on a daily basis, except
for narrow strips between orbits of about 0.5° wide at the equator.
The TROPOMI operational HCHO and $NO_2$ retrieval algorithms have been fully described in De Smedt (2022)
and Van Geffen et al. (2022b), respectively. The first common step is to derive slant columns by performing a
spectral fit using the Differential Optical Absorption Spectroscopy (DOAS) method. Specifics for the SCD
retrieval are provided in Table S1. After the DOAS spectral fitting, a two-step normalization of the HCHO slant
columns is performed to remove any remaining global offset and possible stripes. Then the corrected differential
SCDs (dSCDs) are converted to vertical columns using AMFs at 340 nm. The AMFs are derived from a pre-
calculated look-up table (LUT) storing altitude-dependent AMFs calculated with the VLIDORT v2.6 radiative
transfer model. This approach implements implicit aerosol corrections by assuming that aerosols can be simply
treated as "effective clouds", and uses the OMI-based monthly MLER dataset for surface reflectance. The HCHO
vertical profile shape is specified from TM5-MP daily analyses. For pixels with partly cloudy scenes, a cloud
correction is applied based on the independent pixel approximation (IPA) (Martin et al., 2002), using cloud
fraction (CF), cloud top pressure (CP) and cloud albedo information from the OCRA/ROCINN-CRB product:

$$M = w \cdot M_{\text{cld}} + (1 - w) \cdot M_{\text{clr}} \tag{1}$$

In Eq. (1), $w$ is the cloud radiance fraction (CRF), $M_{\text{cld}}$ the cloudy-sky AMF and $M_{\text{clr}}$ the clear-sky AMF. In the
OCRA/ROCINN-CRB cloud retrieval, OCRA first computes the cloud fraction using a broad-band UV/VIS color-
space approach with two colors: Green (405–495 nm) and Blue (350–395 nm); then ROCINN-CRB calculates the
cloud height and cloud albedo using in and around the oxygen ($O_2$) A-band (~760 nm). In the final step, TM5-MP
HCHO vertical columns in the reference region are added as the compensation for the background HCHO from
methane ($CH_4$) oxidation in the equatorial Pacific. The final tropospheric HCHO VCD, $N_V$, can be written as
follows:

$$N_V = \frac{N_S - N_{S,0}}{M} + \frac{M_{clear,0}}{M} N_{V,0}^{\text{TM5}-\text{MP}} \tag{2}$$

with $(N_S - N_{S,0})$ being the corrected HCHO differential slant column, $M$ the HCHO AMF, $M_{clear,0}$ the HCHO clear-
sky AMF in the reference region ([90ºS, 90ºN], [180ºW, 120ºW]), and $N_{V,0}^{\text{TM5}-\text{MP}}$ the HCHO vertical column from
a daily latitude-dependent polynomial, which is fitted through 5° latitude bin means of TM5-MP HCHO vertical
columns in the reference region (De Smedt, 2022).
For $NO_2$, a de-striping is also applied to the fitted slant columns even though the systematic across-track features
are very small (Van Geffen et al., 2020). The second step is the stratosphere-troposphere separation, where TM5-
MP is used to assimilate TROPOMI total $NO_2$ SCDs, determine the stratospheric $NO_2$ SCDs and, by subtraction,
infer the tropospheric $NO_2$ SCDs. To calculate tropospheric $NO_2$ AMFs, the operational algorithm applies implicit
aerosol corrections, uses $NO_2$ a priori profile shapes from TM5-MP daily analyses, and adopts a DLER at 440 nm
from the KNMI TROPOMI DLER v1.0 surface reflectance database. For the cloud correction, it takes the cloud
top pressure from the FRESCO-S product (using the $O_2$ A-band at ~760 nm) and retrieves an effective cloud
fraction (ECF) by fitting the observed continuum reflectance to a simulated reflectance at 440 nm, assuming an
optically thick Lambertian cloud with a fixed cloud albedo of 0.8. The tropospheric $NO_2$ VCD, $N_V^{\text{trop}}$, can be
written as follows:

$$N_V^{\text{trop}} = \frac{N_S^{\text{total}} - N_S^{\text{strat}}}{M} \tag{3}$$

with $(N_S^{\text{total}} - N_S^{\text{strat}})$ the tropospheric $NO_2$ slant column and $M$ the tropospheric $NO_2$ AMF.
**2.2 Improved POMINO-TROPOMI algorithm for global HCHO and $NO_2$ AMF calculations**
Focusing on the improvement of global HCHO and $NO_2$ AMF calculations as well as their consistency, we use an
updated POMINO-TROPOMI parallelized AMFv6 package (Figure S1) driven by the LInearized Discrete
Ordinate Radiative Transfer code (LIDORT) version 3.6 inherited from previous POMINO products (Liu et al.,
2020). The DOAS spectral fit, HCHO dSCD background correction and $NO_2$ stratosphere-troposphere separation
are not included in this study, so corrected HCHO dSCDs and tropospheric $NO_2$ SCDs are directly taken from the
RPRO HCHO v2.4.1 product and RPRO $NO_2$ v2.4.0 product, respectively. Compared to the previous HCHO
v2.3.0 processor, HCHO v2.4.1 processor uses new improved Level 1b v2.1.0 data products as input, and has been
applied for a full mission reprocessing starting from $7^{\text{th}}$ May 2018. For $NO_2$, the improvements of the v2.4.0
processor include the use of a DLER climatology derived from TROPOMI observations and new improved Level
1b v2.1.0 data products as input, which has also been used for a full mission reprocessing from $1^{\text{st}}$ May 2018.
Detailed information of S5P TROPOMI L2 HCHO and $NO_2$ processing baseline, including the processor version,
in-operation period and relevant improvements can be found at https://sentiwiki.copernicus.eu/web/s5p-
processing.
Table1 lists the main improvements in the POMINO AMF algorithm compared to the RPRO algorithms. POMINO
calculates the AMFs with online pixel-by-pixel RT simulations rather than using the LUT. Explicit aerosol
corrections are implemented at the corresponding wavelengths of HCHO and $NO_2$, respectively, based on the
aerosol information from Global Earth Observing System Composition Forecast (GEOS-CF; Keller et al., 2021)
v1.0 and Moderate Resolution Imaging Spectroradiometer (MODIS) satellite data. We convert GEOS-CF vertical
volume mixing ratio profiles to optical depth profiles for each aerosol type, i.e., dust, sulfate-nitrate-ammonium
(SNA), organic carbon (OC), black carbon (BC) and sea salt, by using high-spectral-resolution aerosol optical
parameters from the GEOS-Chem website
(https://ftp.as.harvard.edu/gcgrid/data/aerosol_optics/hi_spectral_res/v9-02/, last access: 23 July 2024). We then
convert component-specific aerosol information to vertical profiles of aerosol extinction coefficient, single

scattering albedo and phase function. We further use monthly aerosol optical depth (AOD) data from MODIS/Aqua Collection 6.1 MYD04_L2 dataset, with spatial and temporal interpolation for missing values, to constrain the model AOD (Lin et al., 2014). Daily a priori HCHO and $NO_2$ profile shapes at TROPOMI overpass time are also obtained from GEOS-CF v1.0 at the spatial resolution of 0.25º × 0.25º. Detailed comparison of the specifications between GEOS-CF and TM5-MP is provided in Table S2.

In $NO_2$ AMF calculations, to account for the surface reflectance anisotropy over lands and coastal ocean regions, we use bidirectional reflectance distribution function (BRDF) coefficients around 470 nm (band 3; bandwidth: 459 – 479 nm) from the MODIS MCD43C2.061 dataset. The reason for the choice of MODIS BRDF over KNMI TROPOMI DLER is that the operational MODIS BRDF algorithm fully characterizes the dependence of surface reflectance on the solar zenith angle (SZA), viewing zenith angle (VZA) and relative azimuth angle (RAA) by a linear combination of an isotropic parameter plus the volumetric and geometric scattering kernels (Roujean et al., 1992; Zhou et al., 2010), while the DLER model only considers the satellite viewing angle (Tilstra et al., 2024). For HCHO, given that the UV spectral band is not included in the MODIS instrument, we decided to use the climatological DLER at 340 nm from the KNMI TROPOMI DLER v2.0 database.

To allow a consistent cloud correction, we use the same cloud information for both HCHO and $NO_2$ AMF calculation. For each pixel, we acquire the cloud parameters by (1) taking the cloud top pressure from the FRESCO-S cloud product, and (2) re-calculating the cloud fraction at 440 nm in a similar way as used in the operational $NO_2$ algorithm. To simulate the TOA reflectance at 440 nm to derive cloud fraction, we use the ancillary parameters fully consistent with those used in $NO_2$ AMF calculation, i.e., a surface reflectance derived from MODIS BRDF coefficients and explicit aerosol information. Previous studies have demonstrated that in most cases, explicit aerosol corrections lead to reduced cloud (radiance) fractions, especially over regions with heavy aerosol loads such as the North China Plain in winter (Lin et al., 2015); while over regions where frequent aerosol-cloud overlap occurs such as Southeast China in spring, the explicit corrections for absorbing aerosols overlying the cloud deck lead to increased cloud fraction (Jethva et al., 2018). Such differences are because the optical effects of aerosols are separated from those of clouds.

Based on the POMINO structure, we implemented a series of sensitivity tests to assess the importance of structural uncertainties that arise when different ancillary parameters or methodologies are applied to the same data. For HCHO, we first conducted the test "Fst_ORcp" (Case F1) by (1) re-calculating the cloud fraction at 340 nm based on the reflectance derived using TROPOMI L1B radiance dataset version 2.1 in TROPOMI spectral band 3 (305-400 nm), and irradiance dataset version 2.1 for the Ultra-violet, Visible and Near-Infrared (UVN) module post-processed by BIRA-IASB, and (2) using the cloud top pressure from OCRA/ROCINN-CRB product. Therefore, the differences between POMINO HCHO columns (Case F0) and those of the test "Fst_ORcp" represent the structural uncertainty from the cloud correction using different cloud products. Based on the test "Fst_ORcp", we separately evaluate the effect of aerosol correction, surface reflectance and a priori profile shapes by conducting the tests "Fst_imaer" (Case F2), "Fst_mler" (Case F3) and "Fst_tm5" (Case F4), respectively. Note that in all sensitivity tests, only HCHO AMFs are changed accordingly, while we keep using GEOS-CF HCHO columns for background correction.

Similarly, for $NO_2$ AMF calculations, based on POMINO $NO_2$ retrievals as the reference (Case N0), tests "Nst_imaer" (Case N1), "Nst_dler" (Case N2) and "Nst_tm5" (Case N3) are used to quantify the individual effect of aerosol correction, surface reflectance and a priori profile shapes. However, we noticed that the $NO_2$ differences

between POMINO and RPRO products can hardly be explained by the linear combination of the individual effect
of each ancillary parameter as in the HCHO analysis. Therefore, we further conducted an additional test "Nst_joint"
(Case N4) to "mimic" the AMF calculation in the RPRO algorithm, quantifying the joint effect of implicit aerosol
corrections, KNMI TROPOMI DLER and TM5-MP a priori NO$_2$ profile shapes.
**Table 1.** Comparison of ancillary parameters between POMINO and RPRO operational products, and sensitivity tests on the
corresponding ancillary parameters ("S.A.P." means "Same as POMINO").

| Species | Product or sensitivity test case | RT model | Aerosol correction | Surface reflectance | Cloud correction | A priori profiles |
|---|---|---|---|---|---|---|
| HCHO | RPRO v2.4.1 | VLIDORT v2.6 (LUT) | Implicit | OMI-based monthly MLER at 340 nm | CF and CP: OCRA/ROCINN-CRB | TM5-MP (1º × 1º) |
| | POMINO (Case F0) | LIDORT v3.6 (online) | Explicit | KNMI TROPOMI v2.0 DLER at 340 nm[(1)] | CF and CP: same as POMINO NO$_2$ | GEOS-CF (0.25º × 0.25º) |
| | Fst_ORcp (Case F1) | S.A.P. | S.A.P. | S.A.P. | CF: calculated at 340 nm CP: OCRA/ROCINN-CRB | S.A.P. |
| | Fst_imaer (Case F2) | S.A.P. | Implicit | S.A.P. | CF: re-calculated at 340 nm[(2)] CP: OCRA/ROCINN-CRB | S.A.P. |
| | Fst_mler (Case F3) | S.A.P. | S.A.P. | KNMI TROPOMI v2.0 MLER at 340 nm[(1)] | CF: re-calculated at 340 nm[(3)] CP: OCRA/ROCINN-CRB | S.A.P. |
| | Fst_tm5 (Case F4) | S.A.P. | S.A.P. | S.A.P. | CF: calculated at 340 nm CP: OCRA/ROCINN-CRB | TM5-MP (1º × 1º) |

(1) KNMI TROPOMI v2.0 DLER at 340 nm over lands and coastal ocean regions, and MLER at 340 nm over open oceans.

(2) Fst_imaer (Case F2) cloud fraction is re-calculated with implicit aerosol corrections and different from that of Case F1.

(3) Fst_mler (Case F3) cloud fraction is re-calculated with KNMI TROPOMI v2.0 MLER and different from that of Case F1.

| Species | Product or sensitivity test case | RT model | Aerosol correction | Surface reflectance | Cloud correction | A priori profiles |
|---|---|---|---|---|---|---|
| NO$_2$ | RPRO v2.4.0 | DAK v3.2 (LUT) | Implicit | KNMI TROPOMI v1.0 DLER at 440 nm | CF: calculated at 440 nm CP: FRESCO-S | TM5-MP (1º × 1º) |
| | POMINO (Case N0) | LIDORT v3.6 (online) | Explicit | MODIS MCD43C2.061 BRDF around 470 nm[(4)] | CF: re-calculated at 440 nm CP: FRESCO-S | GEOS-CF (0.25º × 0.25º) |
| | Nst_imaer (Case N1) | S.A.P. | Implicit | S.A.P. | CF: re-calculated at 440 nm[(6)] CP: FRESCO-S | S.A.P. |
| | Nst_dler (Case N2) | S.A.P. | S.A.P. | KNMI TROPOMI v2.0 DLER at 440 nm[(5)] | CF: re-calculated at 440 nm[(7)] CP: FRESCO-S | S.A.P. |
| | Nst_tm5 (Case N3) | S.A.P. | S.A.P. | S.A.P. | S.A.P. | TM5-MP (1º × 1º) |
| | Nst_joint (Case N4) | S.A.P. | Implicit | KNMI TROPOMI v2.0 DLER at 440 nm[(5)] | CF: re-calculated at 440 nm[(8)] CP: FRESCO-S | TM5-MP (1º × 1º) |

(4) MODIS MCD43C2.061 BRDF around 470 nm over lands and coastal ocean regions, and KNMI TROPOMI v2.0 MLER at 440 nm over open oceans.

(5) KNMI TROPOMI v2.0 DLER at 440 nm over lands and coastal ocean regions, and MLER at 440 nm over open oceans.

(6) Nst_imaer (Case N1) cloud fraction is re-calculated with implicit aerosol corrections and different from that of Case N0.

(7) Nst_dler (Case N2) cloud fraction is re-calculated with KNMI TROPOMI v2.0 DLER and different from that of Case N0.

(8) Nst_joint (Case N4) cloud fraction is re-calculated with implicit aerosol corrections and KNMI TROPOMI v2.0 DLER, and different from that of Case N0.


## 2.3 Ground-based MAX-DOAS datasets

Ground-based MAX-DOAS instruments can provide vertical columns and profiles of trace gases from the surface
up to the lower free troposphere (around 4 km). The measurement sensitivity is the highest near the surface and
decreases at higher altitudes. Information on ground-based MAX-DOAS measurements used in this study is
summarized in Table 2 with locations specified in Figure S2. For each site, we use Fiducial Reference
Measurements for Ground-based DOAS Air-Quality Observations (FRM4DOAS; https://frm4doas.aeronomie.be/,
Van Roozendael et al., 2024) version 01.01 harmonized HCHO and NO$_2$ data if available, otherwise we use data
generated by principal investigators of each instrument using non-harmonized retrieval settings. The aim of the
FRM4DOAS project is to minimize inhomogeneities in the current MAX-DOAS network to provide reference

datasets for satellite data validation. So far, many MAX-DOAS sites have been used for validation (De Smedt et al., 2021; Pinardi et al., 2020; Verhoelst et al., 2021; Yombo Phaka et al., 2023), but this is the starting point of the FRM$_4$DOAS project and much more sites will join the centralized processing facility.

According to previous studies, the total estimated uncertainty of ground-based MAX-DOAS measurements in polluted conditions is about 30% for HCHO and NO$_2$ VCDs (De Smedt et al., 2021; Verhoelst et al., 2021). The mean bias is due mainly to systematic uncertainties related to AMF calculations. The uncertainty may also vary when different report strategies are used. Routine validation results show an overall bias of −37% for HCHO and −28% for NO$_2$ in the operational TROPOMI products compared to MAX-DOAS measurements in the validation report (available at https://mpc-vdaf.tropomi.eu/).

**Table 2.** MAX-DOAS datasets used for the validation. The sites are listed in alphabetical order based on the first letter of the site name.

| Station, country (lat/long) | Species | Owner/group | Retrieval type | Reference |
|---|---|---|---|---|
| Athens, Greece (38.05ºN, 23.86ºE) | NO$_2$ | IUPB[1] | FRM$_4$DOAS 01.01 | https://frm4doas.aeronomie.be/ Van Roozendael et al. (2024) |
| Bremen, German (53.10ºN, 8.85ºE) | HCHO and NO$_2$ | IUPB | FRM$_4$DOAS 01.01 | https://frm4doas.aeronomie.be/ Van Roozendael et al. (2024) |
| Cabauw, the Netherlands (51.97ºN, 4.93ºE) | HCHO and NO$_2$ | KNMI[2] | FRM$_4$DOAS 01.01 | https://frm4doas.aeronomie.be/ Van Roozendael et al. (2024) |
| Cape Hedo, Japan (26.87ºN, 128.25ºE) | NO$_2$ | JAMSTEC[3] | Parameterized profiling (PP) | Kanaya et al. (2014) |
| Chiba, Japan (35.63ºN, 140.10ºE) | NO$_2$ | ChibaU[4] | Parameterized profiling (PP) | Irie et al. (2011, 2012, 2015) |
| De Bilt, the Netherlands (52.10ºN, 5.18ºE) | HCHO and NO$_2$ | KNMI | FRM$_4$DOAS 01.01 | https://frm4doas.aeronomie.be/ Van Roozendael et al. (2024) |
| Fukue, Japan (32.75ºN, 128.68ºE) | NO$_2$ | JAMSTEC | Parameterized profiling (PP) | Kanaya et al. (2014) |
| Kinshasa, Democratic Republic of Congo (4.3ºS, 15.30ºE) | HCHO and NO$_2$ | BIRA-IASB[5] | FRM$_4$DOAS 01.01 | https://frm4doas.aeronomie.be/ Van Roozendael et al. (2024) |
| Mohali, India (30.67ºN, 76.74ºE) | HCHO and NO$_2$ | IISER[6]/MPIC[7] | QA4ECV harmonization procedure | De Smedt et al. (2021); Kumar et al. (2020) |
| Xianghe, China (39.75ºN, 116.96ºE) | HCHO and NO$_2$ | BIRA-IASB | FRM$_4$DOAS 01.01 | https://frm4doas.aeronomie.be/ Van Roozendael et al. (2024) |
| Yokosuka, Japan (35.32ºN, 139.65ºE) | NO$_2$ | JAMSTEC | Parameterized profiling (PP) | Kanaya et al. (2014) |

(1) Institute of Environmental Physics, University of Bremen
(2) Royal Netherlands Meteorological Institute
(3) Japan Agency for Marine-Earth Science and Technology
(4) Chiba University
(5) Royal Belgian Institute for Space Aeronomy
(6) Indian Institute of Science Education and Research
(7) Max Planck Institute for Chemistry

## 2.4 PGN/Pandora datasets

The Pandonia Global Network (PGN) is a large-scale global network providing ground-based observations of multiple atmospheric reactive trace gases, including HCHO and NO$_2$, and associated uncertainty values for satellite validation and other scientific activities. It is based on ground-based passive spectrometer systems called "Pandora" that can perform sun, moon and sky observations. The datasets have been widely used to validate HCHO and NO$_2$ measurements from satellite instruments and field campaigns (Herman et al., 2019; Kai-Sikhakhane et al., 2024; Li et al., 2021; Liu et al., 2024a; Verhoelst et al., 2021).

Herman et al. (2009) reported that the nominal estimated uncertainty of total $NO_2$ columns is $0.27 \times 10^{15}$ molec.cm$^{-2}$ for the random part and $2.7 \times 10^{15}$ molec.cm$^{-2}$ for the systematic part, and an uncertainty of 20% is reported by comparisons with in-situ measurements (Verhoelst et al., 2021). However, the newer PGN $NO_2$ rnvs3p1-8 data, which are employed in this study, have considerably lower uncertainties due to changes in (1) the optical setup, (2) the gas-calibration approach and (3) a more accurate $NO_2$ effective temperature estimation. As reported in the PGN data products Readme (https://publications.pandonia-global-network.org/manuals/PGN_DataProducts_Readme.pdf), the combined uncertainty increases with decreasing SZA, reaching $\sim 0.45 \times 10^{15}$ molec.cm$^{-2}$ for $NO_2$ rnvs3p1-8 data and $\sim 1.2 \times 10^{15}$ molec.cm$^{-2}$ for HCHO rfus5p1-8 data at SZA=10° (median uncertainty over 137 data sets). The report uncertainty does not yet include the impact of spectral fitting quality and is therefore a lower limit. This uncertainty component will be included in a future PGN release; at Izana site, it is estimated to increase the reported uncertainty at SZA=10° to $1.0 \times 10^{15}$ molec.cm$^{-2}$ for $NO_2$ and $3.0 \times 10^{15}$ molec.cm$^{-2}$ for HCHO.

In this work, we use HCHO rfus5p1-8 and $NO_2$ rnvs3p1-8 direct sun total column measurements only from the ESA Validation Data Centre (EVDC) (https://evdc.esa.int, last access: 17 July 2024), because the PGN sub-dataset submitted to EVDC undergoes a more thorough quality check, in which the issues in PGN HCHO retrievals are mostly mitigated. A total of 22 sites across the globe have valid measurements for HCHO and $NO_2$ validation in the period of study (Figure S2).

**2.5 Data use and validation statistics**

For comparison between satellite HCHO data, we filter out the retrieved data based on the following criteria: we exclude pixels with RPRO quality assurance values (QA) ≤ 0.5, which includes SZA or VZA > 70° or activated snow/ice flag. We also exclude pixels with POMINO-derived CRFs at 440 nm greater than 0.5, to minimize the impact of cloud contamination. The same criteria are applied to the $NO_2$ comparison as well. To examine the spatial distribution, gridded tropospheric HCHO and $NO_2$ VCDs in April, July, October 2021, and January 2022 at a resolution of 0.25° × 0.25° are calculated using an area-weighted oversampling technique (Zhang et al., 2023). For comparisons between satellite and ground-based HCHO data, we take two successive steps for data processing. First, we calculate the daily average HCHO columns from ground-based MAX-DOAS and PGN measurements within the time window between 11:00 and 16:00 LT. For PGN data, we only use those with the flag "assured high quality" (data quality flag of 0) or "not-assured high quality" (data quality flag of 10) ((https://www.pandonia-global-network.org/wp-content/uploads/2024/11/PGN_DataProducts_Readme_v1-8-9.pdf). Then we calculate daily average satellite HCHO columns based on pixels selected using the cloud information from POMINO retrieval, with the pixel center located within a radius of 20 km to the instruments. The daily collocated data pair is considered valid only if 10 satellite pixels or more are used for calculation. The processing for $NO_2$ data is different from that of HCHO in three aspects: (1) the time window for $NO_2$ is between 13:00 to 14:00 LT, as the diurnal variation of $NO_2$ is much stronger than that of HCHO; (2) the radius between the satellite pixel center and the instrument is 5 km, considering the much larger spatial gradient of the $NO_2$ distribution and less noise in the $NO_2$ retrieval; (3) we derive PGN tropospheric $NO_2$ columns each day by subtracting stratospheric $NO_2$ columns from the RPRO $NO_2$ v2.4.0 L2 product over the instrument from the total $NO_2$ columns, in order to make them comparable with satellite tropospheric $NO_2$ columns (Pinardi et al., 2020). Based on collocated HCHO and $NO_2$ columns, we further compare the daily tropospheric column ratio of

formaldehyde to nitrogen dioxide (FNR) derived from satellite products and ground-based MAX-DOAS and PGN
measurements.
To quantify the performance of satellite products relative to ground-based measurements, we derive slope, offset
and correlation of the linear regression using the robust Theil-Sen estimator (Sen, 1968), which is insensitive to
occasional outliers. In a relative sense, we use normalized mean bias (NMB) to quantify the deviation between
satellite and ground-based measurements:
$$\text{NMB} = \frac{\overline{\Omega^{\text{SAT}}} - \overline{\Omega^{\text{ground-based}}}}{\overline{\Omega^{\text{ground-based}}}} \times 100\% \tag{4}$$

with $\Omega$ being the HCHO or $NO_2$ vertical column in Sects. 6.1 and 6.2, and FNR in Sect. 6.3.
**3 Comparison of HCHO and NO₂ columns between POMINO and RPRO products**
Figures 1a and c illustrate the global distribution of tropospheric HCHO VCDs averaged over April, July, October
2021 and January 2022 from POMINO and RPRO retrieval, respectively. High levels of tropospheric HCHO
columns ($> 10 \times 10^{15}$ molec.cm$^{-2}$) are evident over the Amazonia Rainforest, Sub-Saharan Africa, South and East
Asia as well as North Australia. Enhanced HCHO concentrations are also noticeable in the southeastern United
States of America (USA) and Mexico, while localized hotspots with lower magnitudes are evident in the Middle
East and Europe. Over the remote background regions, HCHO is primarily from $CH_4$ oxidation, and the abundance
is about $3 \times 10^{15}$ molec.cm$^{-2}$ at maximum. Similarly, Figs. 1b and d show the POMINO and RPRO tropospheric
$NO_2$ VCDs in April, July, October 2021 and January 2022. High $NO_2$ columns are visible over three well-known
polluted regions, i.e., North China Plain, West Europe, and East USA, with strong hotspot signals over megacities
and metropolitan areas across the globe. Low $NO_2$ content in the remote atmosphere comes from aviation and
ship emissions, natural biogenic emissions, lightning and oxidation of long-lifetime species such as peroxyacetyl
nitrate (PAN).

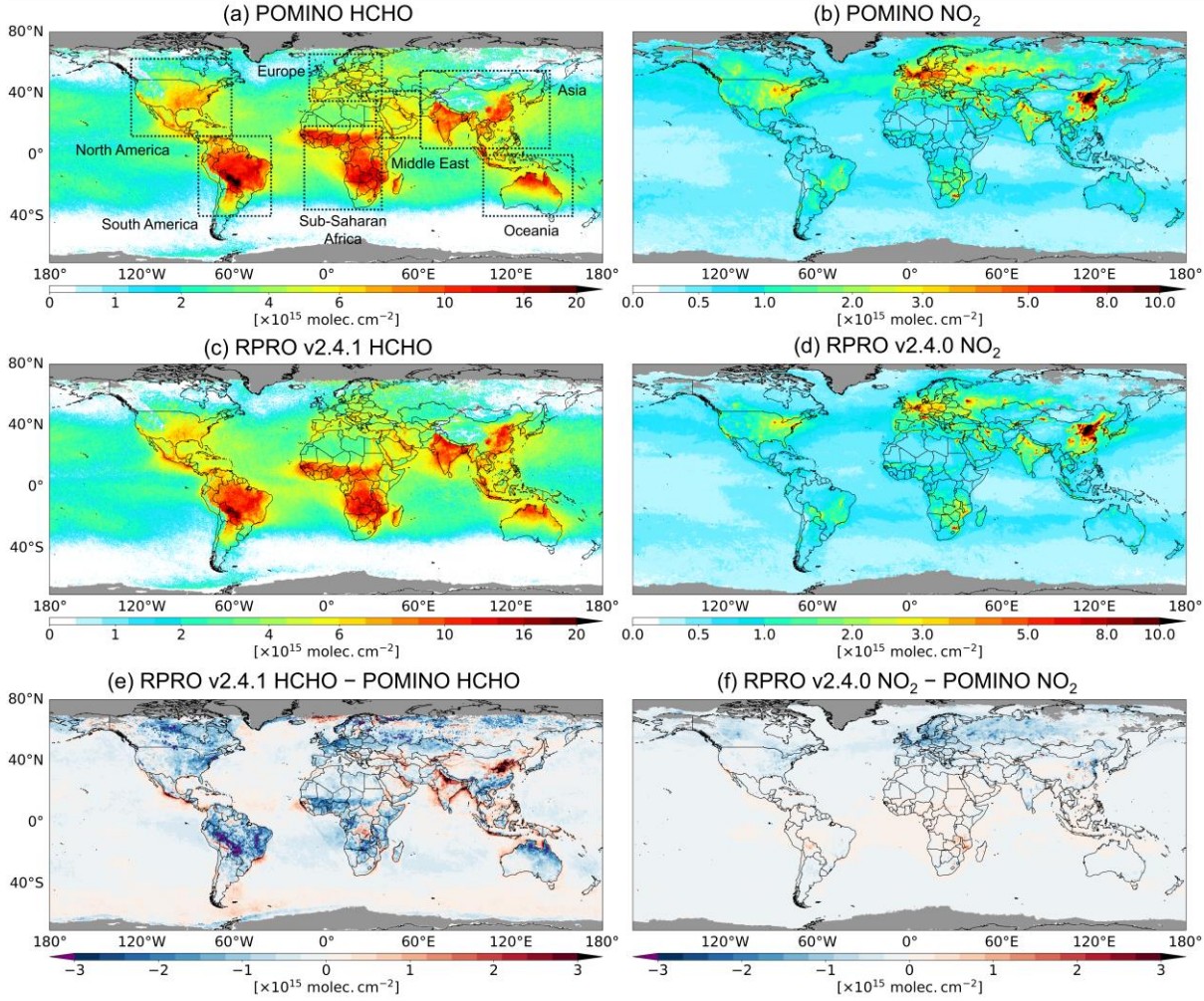

**Figure 1.** Spatial distribution of POMINO tropospheric HCHO and NO$_2$ VCDs (**an** and **b**), RPRO tropospheric HCHO and NO$_2$ VCDs (**c** and **d**), and respective absolute differences (**e** and **f**) at a spatial resolution of 0.25º × 0.25º averaged in April, July, October 2021, and January 2022. The black dashed rectangles illustrate the spatial range of the regions used for comparison. The regions in gray mean that there are no valid observations.

A high qualitative agreement is observed for both HCHO and NO$_2$ VCDs between RPRO and POMINO retrievals, as the same HCHO dSCDs and tropospheric NO$_2$ SCDs are used. However, as shown in Fig. 1e, RPRO HCHO tropospheric columns are lower by $2 \times 10^{15}$ molec.cm$^{-2}$ or more over almost all regions with elevated HCHO columns except North India and North China Plain; RPRO NO$_2$ columns are also lower than those of POMINO over most East China, India, Europe, and North America by up to about 20% in a relative sense, despite the positive differences over Sub-Saharan Africa and some cities such as Xi'an, Teheran, and Los Angeles (Fig. 1f). We further make the comparison in seven specific regions (bounded by black rectangles in Fig. 1a): North America (125ºW-60ºW, 10ºN-65ºN), South America (85ºW-35ºW, 40ºS-10ºN), Europe (10ºW-35ºE, 35ºN-60ºN), Sub-Saharan Africa (15ºW-35ºE, 35ºS-20ºN), Middle East (30ºE-60ºE, 10ºN-40ºN), Asia (60ºE-145ºE, 5ºN-55ºN), and Oceania (100ºE-160ºE, 40ºS-0º). Figure 2 shows the comparison results over the most polluted areas in each region, defined as where the POMINO tropospheric HCHO or NO$_2$ VCDs averaged over April, July, October 2021 and January 2022 exceed their 99 percentiles; results for regional mean comparisons are shown in Figure S3. For HCHO, RPRO data are consistently lower than POMINO by around 15% over polluted areas in five regions, although the difference is small over the Middle East and Asia because of the cancellation between

positive and negative differences on the finer spatial scale. For $NO_2$, RPRO is smaller than POMINO by −19.4%
for North America and −23.3% for Europe. Detailed comparisons for each month are shown in Figure S4 and S5.
Overall, POMINO and RPRO HCHO and $NO_2$ retrievals show excellent agreement in a qualitative sense, but the
column values differ by 10% to 20% on average over polluted areas around the world. Such differences result
from the different cloud correction, aerosol correction, surface reflectance and vertical profile shapes used in AMF
calculations, which will be further discussed in Sect. 4.

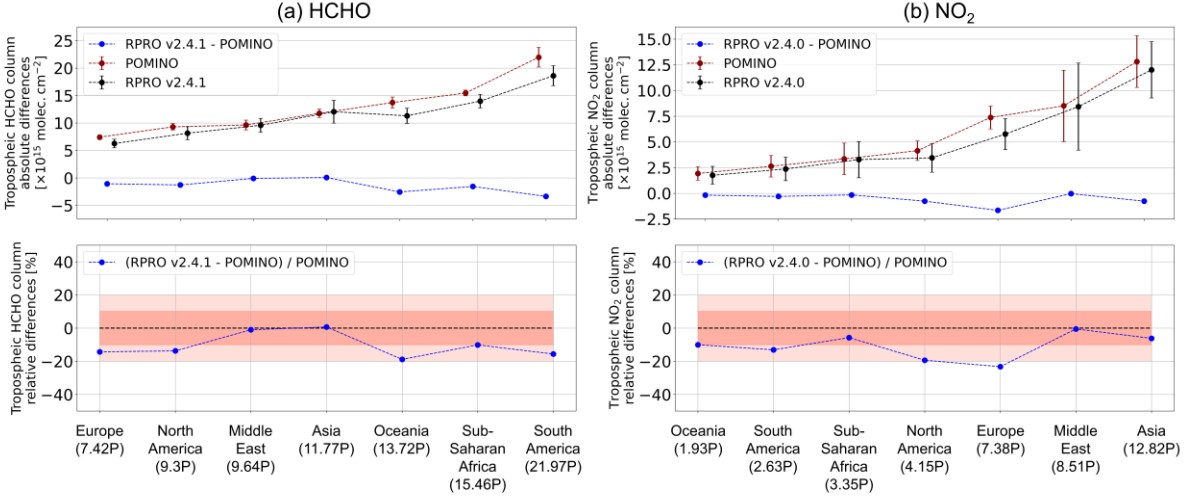

**Figure 2.** Absolute and relative differences between POMINO and RPRO (**a**) HCHO and (**b**) $NO_2$ tropospheric columns
averaged in April, July, October 2021, and January 2022 over polluted areas (defined as where POMINO mean HCHO or $NO_2$
columns exceed their 99 percentiles) in seven regions. Regions are sorted as a function of POMINO mean HCHO or $NO_2$
columns, with values (in the unit of "P" as $Pmolec.cm^{-2} = 1 \times 10^{15}$ molec.cm$^{-2}$) shown in the brackets in the bottom axis. Mean
POMINO (red) and RPRO (black) columns are also plotted with the absolute differences in the upper panel. Error bars
represent the standard deviations of the columns. Pink areas indicate 10% and 20% relative differences.

## 4 Sensitivity tests on AMF input parameters

As listed in Table 1, we implement a series of sensitivity tests to quantify the structural uncertainty from either
individual or joint effect of using different ancillary parameters in the HCHO and $NO_2$ AMF calculation. The time
period selected for the sensitivity analysis is July 2021 and January 2022, representing the summer and winter
time, respectively. Note that one of the most important features of the POMINO HCHO and $NO_2$ retrievals is that
they use the same cloud parameters for consistent cloud correction. Therefore, besides discussing the effect of
cloud correction based on POMINO cloud parameters, we also compare the differences between HCHO columns
retrieved using different cloud parameters, especially the cloud top pressures. The influences of aerosol correction,
surface reflectance, a priori profile shapes and their joint effect are discussed in the subsequent sub-sections.

### 4.1 Cloud correction

### 4.1.1 Effect of cloud correction based on POMINO cloud parameters

When calculating tropospheric AMFs, it is important to account for the influence of clouds on the radiative transfer
process in the atmosphere (Boersma et al., 2011; De Smedt et al., 2021; Lorente et al., 2017; Martin et al., 2002).
Clouds can either enhance or reduce the sensitivity to the trace gas molecules depending on their height relative
to the trace gas layers (the so-called "albedo" or "shielding" effect, respectively). Despite the relatively large
uncertainty of retrieved cloud parameters in near-cloud-free scenario (defined here as CF ≤ 0.1 or CRF ≤ 0.4)

(Richter and Burrows, 2002), most HCHO and $NO_2$ AMF algorithms make use of the IPA method (Sect. 2.1) to
explicitly account for the cloud effect.
Figure 3 shows the differences between clear-sky AMF and total AMF of all pixels with HCHO or $NO_2$ QA > 0.5
in July 2021 and January 2022, based on the FRESCO-S cloud top pressures and POMINO re-calculated cloud
fractions at 440 nm with explicit aerosol corrections. For both HCHO and $NO_2$, the differences between clear-sky
AMF and total AMF are negative when cloud top pressures are higher than 700 hPa, and their magnitudes continue
to increase along with the cloud top pressures. The negative differences can be as large as −30% for HCHO and
−20% for $NO_2$ when the CRFs are in the interval of 0.45 to 0.5 and cloud top pressures are higher than 900 hPa.
This illustrates the "albedo" effect of low clouds by increasing the contribution of photons from near-surface
layers to the ensemble of photons received at the satellite instrument and thus leading to higher total AMF.

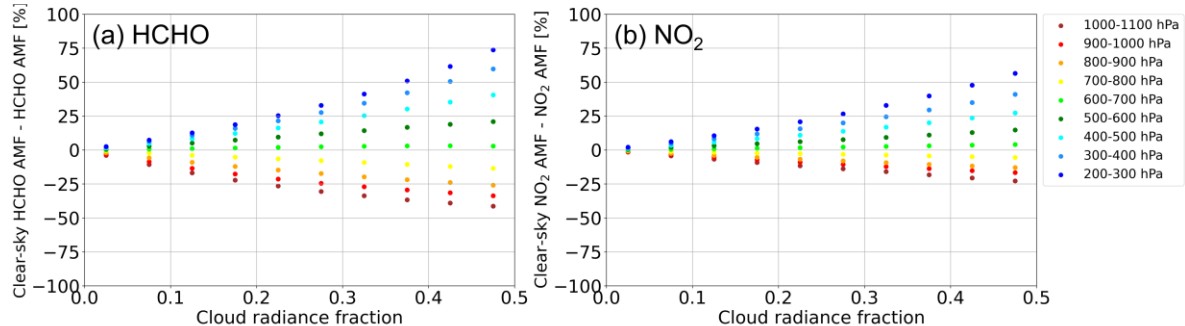


**Figure 3.** Differences of (**a**) HCHO and (**b**) $NO_2$ clear-sky AMF to total AMF for different cloud radiance fraction with an
interval of 0.05 in different cloud top pressure ranges (shown in different colors). All pixels with HCHO or $NO_2$ QA > 0.5 in
July 2021 and January 2022 are included.

On the contrary, clouds with cloud top pressure lower than 700 hPa reflect most photons back to the top of
atmosphere as a "shield" before they reach the HCHO or $NO_2$ abundant layers. As a result, positive differences of
clear-sky AMF to total AMF occur, and they increase as the cloud top pressures decrease, reaching 50% or more
when CRFs are in the interval of 0.4 to 0.5 and cloud top pressures are lower than 400 hPa. This result is also in
line with the previous study by Lorente et al. (2017).
In the global view (Figure 4), for both HCHO and $NO_2$ columns, the difference due to cloud correction (i.e., using
clear-sky AMF versus total AMF) is ±10% on average over high-value regions and can reach 40% over specific
areas. Note that all these comparisons are based on HCHO and $NO_2$ a priori profile shapes from GEOS-CF. The
signs and values of the differences might be different when using the profile shapes from another model, along
with the structural uncertainty discussed in Sect. 4.1.2.
One issue existing in the process of cloud correction in the POMINO retrieval is that only the cloud fraction is re-
calculated with explicit aerosol corrections, while the cloud top pressure is taken from the external dataset, i.e.,
the FRESCO-S cloud product, in which the aerosols are implicitly accounted for. As a result, this step introduces
presumably an aerosol overcorrection issue in the cloud top pressures of partly cloudy pixels, and therefore brings
in additional uncertainties in the AMF calculations. Lin et al. (2015) reported that excluding aerosols leads to an
increase of $O_2$-$O_2$-based cloud top pressures (from 700–900 hPa to 750–950 hPa) over eastern China, but it is
difficult to clarify the mechanism due to its complexity (Lin et al., 2014). Currently there is no direct way to
estimate the effect of aerosol correction on the FRESCO-S cloud height retrieval without doing $O_2$ A-band cloud
retrieval tests, which is beyond the scope of this study. However, below we give an estimation of the uncertainty
in POMINO HCHO and $NO_2$ vertical columns caused by this issue.

Given the fact that, in the retrieval algorithm, the cloud is assumed to be an optically thick Lambertian reflector with a high albedo of 0.8, the cloudy-sky AMF (and hence tropospheric AMF) is very sensitive to the accuracy of the cloud height when the cloud is low and vertically mixed with the aerosols and trace gases. In these cases, we can assume that the retrieved cloud height is primarily influenced by aerosols (Van Geffen et al., 2022a), therefore the aerosol overcorrection issue becomes non-negligible. Focusing on valid pixels for which the difference between the surface pressure and the FRESCO-S cloud top pressure is equal to 100 hPa or less (~17.5% and ~19.9% of total pixels in July 2021 and January 2022, respectively), the aerosol overcorrection uncertainty can be roughly estimated from the difference of HCHO and $NO_2$ vertical columns retrieved using either aerosol-corrected clear-sky AMFs (aerosol correction applied; cloud correction not applied) or aerosol-corrected total AMFs (both aerosol and cloud corrections applied). Based on the results shown in Figure S6, we tentatively estimate the uncertainty to be in the range from 10% to 15% for HCHO, and within 10% for $NO_2$. The estimated $NO_2$ uncertainty level is also supported by the sensitivity test results in Liu et al. (2020). They implemented a "semi-explicit" aerosol correction approach, in which aerosol optical effects are explicitly corrected for clear-sky AMFs, but are excluded for the cloudy-sky portion of partly cloudy pixels, and found the $NO_2$ differences due to the aerosol correction choice for cloudy-sky AMFs vary from 3.1% to 11.2% over eastern China in July 2018. The tentatively estimated uncertainty range above is comparable to or less than that from other ancillary parameters (Sect. 5), and only needs to be taken into account for partly cloudy pixels with low clouds.

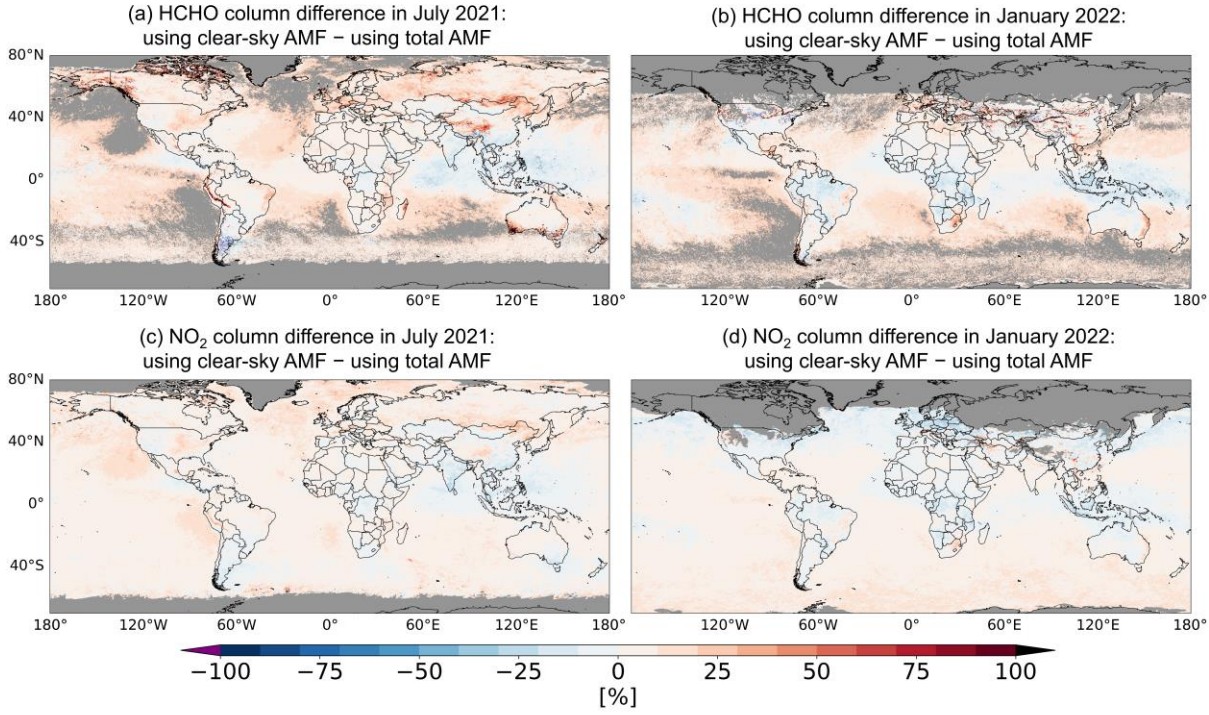

**Figure 4.** Relative differences of tropospheric HCHO (**a** and **b**) and $NO_2$ (**c** and **d**) columns derived using clear-sky POMINO AMF to those using total POMINO AMF in July 2021 and January 2022. The regions in gray mean that there are no valid observations.

### 4.1.2 Structural uncertainty of cloud correction based on different cloud parameters

The structural uncertainty of the cloud correction can be evaluated using cloud parameters from different cloud products. Lorente et al. (2017) have demonstrated that the systematic differences in cloud top pressure can lead to substantial differences in tropospheric $NO_2$ AMFs and VCDs. Focusing on HCHO in this section, we first

compare the effective cloud fractions and cloud top pressures either calculated in different ways or from different products. As shown in the left column of Figure S7, POMINO-based ECF calculated at 440 nm and 340 nm as well as OCRA/ROCINN-CRB ECF show similar global patterns in July 2021. Despite the differences over certain areas, great agreement is exhibited between OCRA/ROCINN-CRB ECF and POMINO-based ECF calculated at 440 nm (linear regression slope of 0.92, offset of 0.02 and correlation coefficient of 0.80), and between POMINO-based ECF calculated at 340 nm and 440 nm (linear regression slope of 0.93, offset of 0.01 and correlation coefficient of 0.93). However, the OCRA/ROCINN-CRB cloud top pressures are significantly higher than those of the FRESCO-S product over the Amazonia Rainforest, Equatorial Africa and East China by 100-300 hPa, while the FRESCO-S cloud top pressures tend to be higher over many other places such as the Intertropical Convergence Zone (ITCZ) over the oceans (Fig. S6f). The comparison results over China are also qualitatively consistent with the findings by Latsch et al. (2022), in which the ROCINN CRB cloud heights differ significantly from those of FRESCO-S when considering low cloud fraction and lowest cloud height values that are critical for tropospheric trace gas retrievals. Such differences are systematic and are caused by different methodologies and ancillary parameters used in each cloud retrieval (Loyola et al., 2018; Van Geffen et al., 2022a), which are also reported in recent validation exercises using independent cloud measurements (Compernolle et al., 2021).

As shown in Fig. 5, by comparing the result of POMINO to the test "Fst_ORcp" (Case F1, using the OCRA/ROCINN-CRB cloud top pressures and the POMINO-based ECFs calculated at 340 nm), we find differences of HCHO columns by up to 20% on average over highly polluted regions, as well as a positive increment over South America. Over remote background regions such as the Pacific Ocean, however, negative differences are found of 0.5-1 × $10^{15}$ molec.cm$^{-2}$. We attribute these differences to different OCRA/ROCINN-CRB and FRESCO-S cloud top pressures, as ECFs in POMINO and Case "Fst_ORcp" are very close. Note that this is a tentative estimate of HCHO column structural uncertainty from the choices of cloud parameters for cloud correction, because the results are dependent on the explicit aerosol corrections and HCHO priori profile shapes used in the tests.

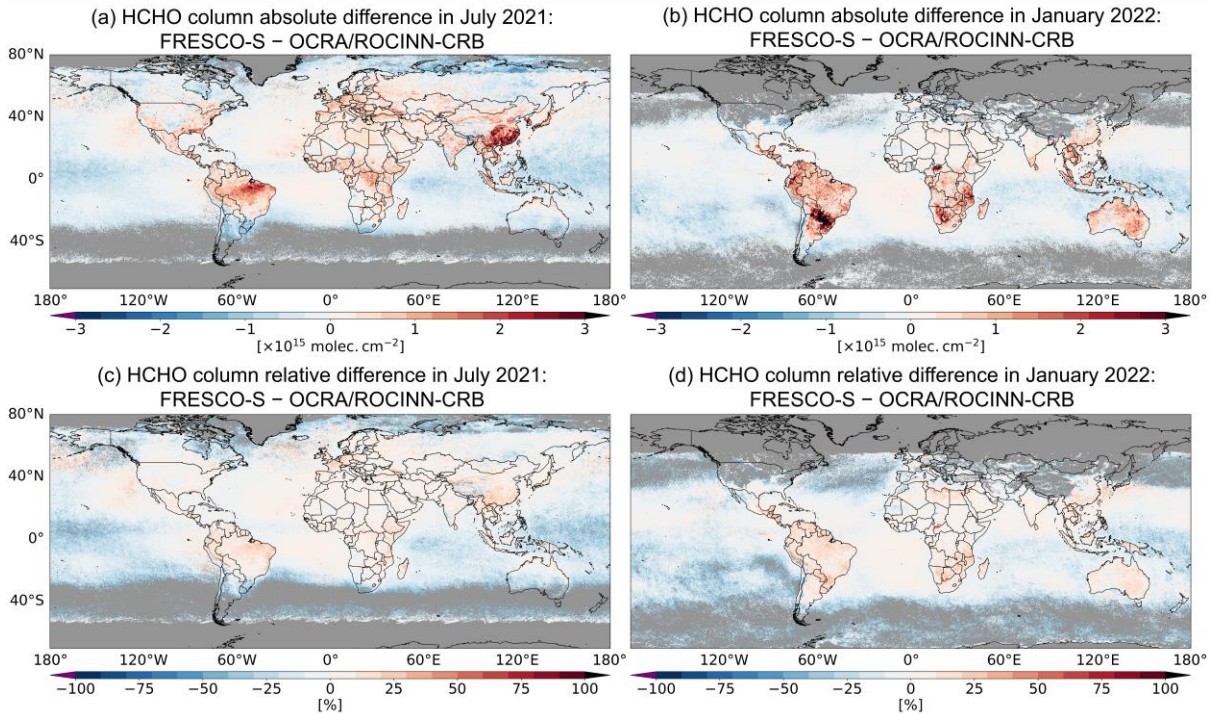

**Figure 5.** Absolute (first row) and relative differences (second row) of tropospheric HCHO columns of POMINO (using FRESCO-S cloud top pressures) to those of the sensitivity test "Fst_ORcp" (using OCRA/ROCINN-CRB cloud top pressures) in July 2021 and January 2022. Different cloud top pressures are emphasized in the title. The regions in gray mean that there are no valid observations.

In summary, the implementation of the cloud correction in HCHO and $NO_2$ retrievals is necessary, and the structural uncertainty due to different cloud parameters needs be taken into consideration in product comparisons. On the other hand, given the different spectral ranges used for trace gas retrievals (HCHO: 340 nm; $NO_2$: 440 nm) and cloud retrievals (OCRA/ROCINN-CRB: $O_2$ A-band between 758 and 771 nm; FRESCO-S: $O_2$ A-band around 760 nm), cloud parameters should always be used with caution, especially for low-cloud-fraction conditions. For example, in the ROCINN-CRB model, priori OCRA cloud fractions smaller than 0.05 are set to zero, and the ROCINN retrieval is not activated under such "clear-sky" conditions. Instead of the NIR spectral range, the $O_2$-$O_2$ cloud algorithm uses the $O_2$-$O_2$ absorption window around 477 nm, but it is more sensitive to low clouds and aerosols. Therefore, further work is still needed to address such discrepancies.

### 4.2 Aerosol correction

The influence of aerosols on AMF calculations is very complicated because they depend on the type of aerosols (scattering or absorbing) and their height relative to the trace gases. The AMFs are generally increased when non-absorbing aerosols are vertically collocated with or lower than the trace gases, while an opposite effect arises when the non-absorbing aerosols reside vertically higher than the trace gases; On the other hand, absorbing aerosols (e.g., black carbon) always reduce the sensitivity of the satellite instruments to the trace gases (Leitão et al., 2010; Lin et al., 2014, 2015; Liu et al., 2024b). Figure S8 shows a global map of AOD at 340 nm and 440 nm used in POMINO retrievals. Areas with heavy aerosol loads in July 2021 include North America, Equatorial Africa, Middle East, India and East China due to biomass burning and/or anthropogenic activities; while in January 2022, the aerosol content is significant in Equatorial Africa, North India and North China Plain. Different aerosol corrections can directly change the clear-sky AMF, affect the retrieval of cloud information (cloud fraction in particular) and modulate the AMF in the cloudy portion of the pixel. The latter two effects influence the total AMF in an indirect way, and the impact on cloud information is often more significant than on cloudy-sky AMF (Vasilkov et al., 2021).

Figure 6 shows that when using clear-sky AMFs to derive vertical columns, implicit aerosol corrections lead to higher HCHO columns by 10% to 20 % over North America in July 2021, and the differences exceed 20% over North India and East China in January 2022. A similar pattern is shown in the $NO_2$ comparison. This is because when aerosols that reside vertically lower than or are mixed with HCHO and $NO_2$ molecules are excluded (i.e., in the case of implicit corrections), the calculated AMFs are lower than those with explicit aerosol corrections. On the other hand, for scenarios with strong anthropogenic emissions or biomass burning, where most HCHO and $NO_2$ molecules are near the surface while aerosols reside above these trace gases, implicit aerosol corrections neglect the strong "shielding" effect of the scattering aerosols and the strong absorption of photons by the absorbing aerosols (e.g., BC), which leads to higher AMFs and lower vertical columns. The negative differences of HCHO columns over the Democratic Republic of Congo in July 2021 (Fig. 6a) can be explained by the second case.

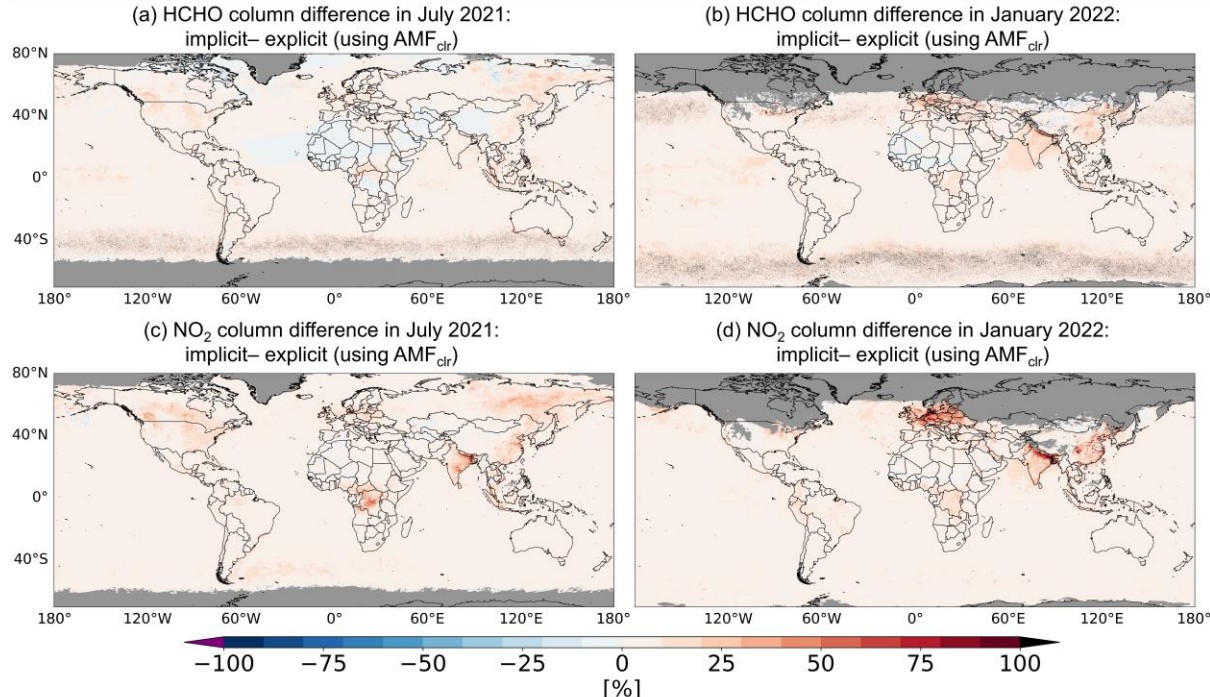

**Figure 6.** Relative differences of tropospheric HCHO (**a** and **b**) and NO$_2$ (**c** and **d**) columns retrieved using clear-sky AMF with implicit aerosol corrections to those with explicit aerosol corrections in July 2021 and January 2022. The regions in gray mean that there are no valid observations.

For cloudy-sky AMF, the impact of non-absorbing aerosols above a cloud is negligible since we assume the cloud to be an optically thick Lambertian reflectivity with a high albedo of 0.8 (Vasilkov et al., 2021). For absorbing aerosols above the clouds, they can reduce the backscattered radiance and hence affect the cloudy-sky AMF. However, Jethva et al. (2018) show that the occurrence of above-cloud absorbing aerosols is most frequent over coastal and oceanic regions because of the long-range transport of aerosols and low-level stratocumulus clouds. Over Southeast Asia during the springtime, the cloudy-sky frequency of occurrence of above-cloud absorbing aerosols is 20% to 40%, probably caused by biomass burning activities. Retrievals under these conditions are mostly discarded because the cloud fractions are too high to meet the filtering criteria for valid pixels (Sect. 2.5). Therefore, the overall influence of implicit aerosol corrections on the cloud-sky AMF can be neglected and the influence on the retrieval of cloud information, especially cloud fraction, is much more significant.

As explained in Sect. 2.2, explicit aerosol corrections affect the retrieved cloud (radiance) fraction due to the inclusion of aerosol radiative contribution. This is also confirmed in Figure S9 that compares retrieved cloud radiance fractions for the implicit versus explicit aerosol correction settings, in both UV and visible bands. As shown in Figure 7, when using cloud-corrected AMFs to consider both direct and indirect aerosol optical effects on the retrieval, the sign of HCHO relative differences over many regions is reversed from positive to negative compared to Figs. 6a and b, such as North and South America. This reflects the enhanced cloud "albedo" effect that increases the calculated HCHO scattering weights over the areas where cloud layers are vertically near or below the HCHO layers. As for NO$_2$, similar results due to enhanced cloud "albedo" effect are found over North America and East Russia in July 2021 (Fig. 7c), but the overall pattern in January 2022 remains the same as that in Fig. 6d. Over the polluted regions in Asia and Europe, implicit aerosol corrections increase the retrieved NO$_2$ columns by 20% to 40% on average. This is because most NO$_2$ molecules over these polluted areas reside within 1 km above the ground and below the FRESCO-S cloud layers during wintertime, so the increased cloud fractions

527  due to implicit aerosol corrections enhance the "shielding" effect on tropospheric $NO_2$ AMF calculation and hence

528  higher $NO_2$ columns. The signs of the HCHO and $NO_2$ differences over North China Plain are not the same,

529  probably because of the differences between HCHO and $NO_2$ vertical profile shapes.

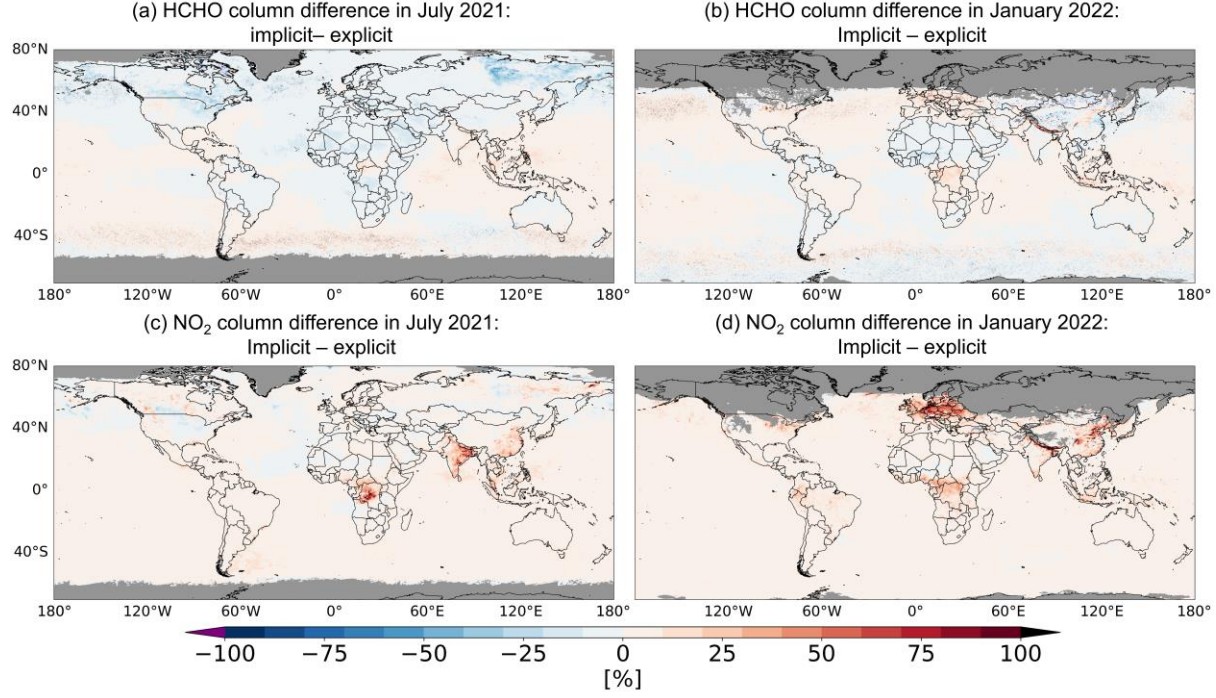

530

**Figure 7.** Relative differences of tropospheric HCHO (**a** and **b**) and $NO_2$ (**c** and **d**) columns retrieved using cloud-corrected total AMF with implicit aerosol corrections (Cases "Fst_imaer" and "Nst_imaer") to those with explicit aerosol corrections (Case "Fst_ORcp" and POMINO $NO_2$) in July 2021 and January 2022. The regions in gray mean that there are no valid observations.

### 4.3 Surface reflectance

Compared to the LER model, which simply assumes the surface to be a Lambertian reflector, DLER partly
accounts for the anisotropy of the surface reflectance by building a certain relationship between the reflectance
and the satellite VZA, but its dependence on the SZA and RAA is still not included. The BRDF model fully
considers the surface optical property as a function of SZA, VZA, RAA and wavelength. At 340 nm, the
directionality of the surface reflectance is small over most regions (Kleipool et al., 2008). Figure S10 compares
the MODIS BRDF-derived blue-sky albedo (BSA, Schaepman-Strub et al., 2006) around 470 nm and KNMI
TROPOMI DLER at 440 nm over lands and coastal ocean regions. In both months, DLER shows higher values
than MODIS BSA except over desert and mountain regions, and the positive differences are larger than 0.1 over
India in July 2021 and East Europe in January 2022. Reasons for these differences are not clear yet, but they are
likely associated with different parameters and corrections for aerosols and snow/ice cover in the algorithm. The
accuracy of the MODIS operational BRDF/albedo product (MCD43) is estimated by 5% to 10% of the field data
at most validation sites studied so far (https://modis-land.gsfc.nasa.gov/ValStatus.php?ProductID=MOD43).
Chong et al. (2024) also provide an estimation of random uncertainties in MODIS MCD43C1 surface reflectances
for various surface types, which vary in the range of 0.01 to 0.03 for most cases.
Figures 8a and b present the influence of surface reflectance on HCHO retrievals. As it is well known that the
directionality of surface reflectance plays a marginal role in the retrieval based on the UV band, nearly no
difference is shown between HCHO columns retrieved using KNMI TROPOMI DLER and MLER at 340 nm.

However, the systematic differences between different MLER products are a more important source of the structural uncertainty in HCHO AMFs. For example, KNMI TROPOMI MLER albedo at 340 nm is found to be consistently lower than OMI climatology monthly MLER albedo used in the RPRO product by 0.01–0.05 (Kleipool et al., 2008; Tilstra et al., 2024).

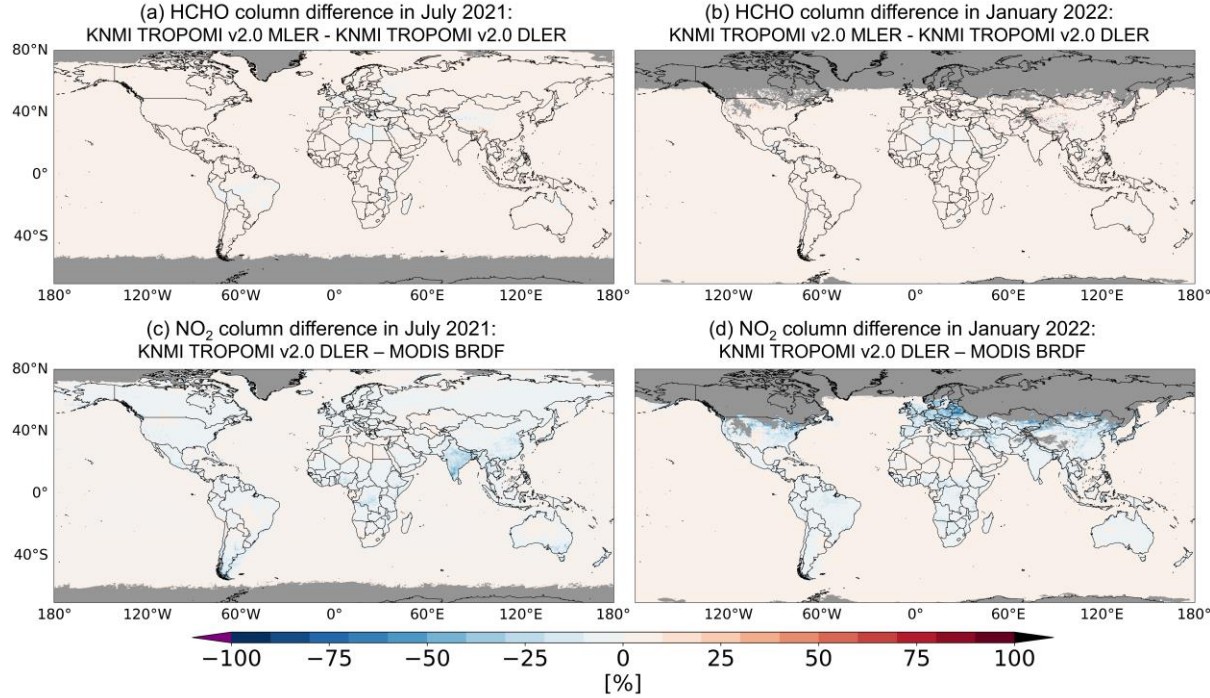

**Figure 8.** Relative differences of tropospheric HCHO columns retrieved using KNMI TROPOMI v2.0 MLER at 340 nm (Case "Fst_mler") to those using KNMI TROPOMI v2.0 DLER at 340 nm (Case "Fst_ORcp") (**a** and **b**), and relative differences of tropospheric NO$_2$ columns retrieved using KNMI TROPOMI v2.0 DLER at 440 nm (Case "Nst_dler") to those using MODIS BRDF at 440 nm (POMINO NO$_2$) (**c** and **d**) in July 2021 and January 2022. The regions in gray mean that there are no valid observations.

As for NO$_2$, Figs. 8c and d show significantly lower tropospheric NO$_2$ VCDs in the test "Nst_dler" (Case N2) than those in the reference POMINO retrieval (Case N0) over most land areas. In January 2022, the NO$_2$ columns retrieved using KNMI TROPOMI DLER are lower by 30% on average over the polluted regions with NO$_2$ columns larger than $10 \times 10^{15}$ molec.cm$^{-2}$ in Europe and North America. Like aerosols, the influence of surface reflectance on AMFs is also a combination of the direct effect on clear-sky AMF and the indirect effect through cloud correction (Boersma et al., 2011). As discussed by Tilstra (2024), DLER should not be considered as the optimal replacement for the BRDF in the VIS wavelength. If the directional surface reflection can be modelled in the RT calculation, it is better to use BRDF to derive surface reflectance for tropospheric NO$_2$ AMF calculation.

**4.4 A priori profiles**

In POMINO, we consistently use GEOS-CF HCHO and NO$_2$ vertical profile shapes as the prior information for AMF calculations. Compared with TM5-MP model of which the spatial resolution is 1º × 1º, GEOS-CF features a much finer spatial resolution (0.25º × 0.25º). The horizontal distributions of GEOS-CF and TM5-MP tropospheric HCHO and NO$_2$ VCDs are shown in Figure S11, and comparisons of monthly mean HCHO and NO$_2$ vertical profile shapes between the models and the ground-based MAX-DOAS measurements are shown in Figure S12. The collocation of model profiles and MAX-DOAS profiles follows the same methodology as described in Sect. 2.5. The differences between GEOS-CF, TM5-MP and MAX-DOAS profiles reflect the imperfections in

these data yet to be fully characterized (Keller et al., 2021; Williams et al., 2017), and they are also an important
source of structural uncertainty in HCHO and $NO_2$ retrievals.
Figure 9 shows the differences in retrieved HCHO and $NO_2$ VCDs caused by using different a priori vertical
profile shapes. The HCHO and $NO_2$ columns retrieved with TM5-MP prior information are obtained using AMFs
re-calculated by combining interpolated POMINO averaging kernels (AK) and TM5-MP a priori profile shapes.
As shown in Figs. 9a and b, the spatial patterns of HCHO relative differences are variable over different places
and in different months, and are generally more significant than the individual effects of clouds, aerosols and
surface reflectance changes (Figs. 4, 7 and 8). At the regional level, the HCHO structural uncertainty from a priori
profile shapes is 20% to 30% over the background clean areas, and 10% to 20% over the polluted areas. In contrast,
the $NO_2$ differences caused by different a priori profile shapes are around 10% over the clean areas and reach 30%
or more over the polluted areas. Over East China, India and the Middle East, localized differences over cities and
polluted regions are obvious (Figs. 9c and d), reflecting the significant differences between TM5-MP and GEOS-
CF $NO_2$ profile shapes. Besides, distinctive patterns along the coastal lines are visible, especially in the HCHO
relative differences. This is caused by the relatively coarse horizontal resolution of TM5-MP, in which the large
heterogeneity of HCHO vertical distribution is smoothed in the 1º × 1º grid.

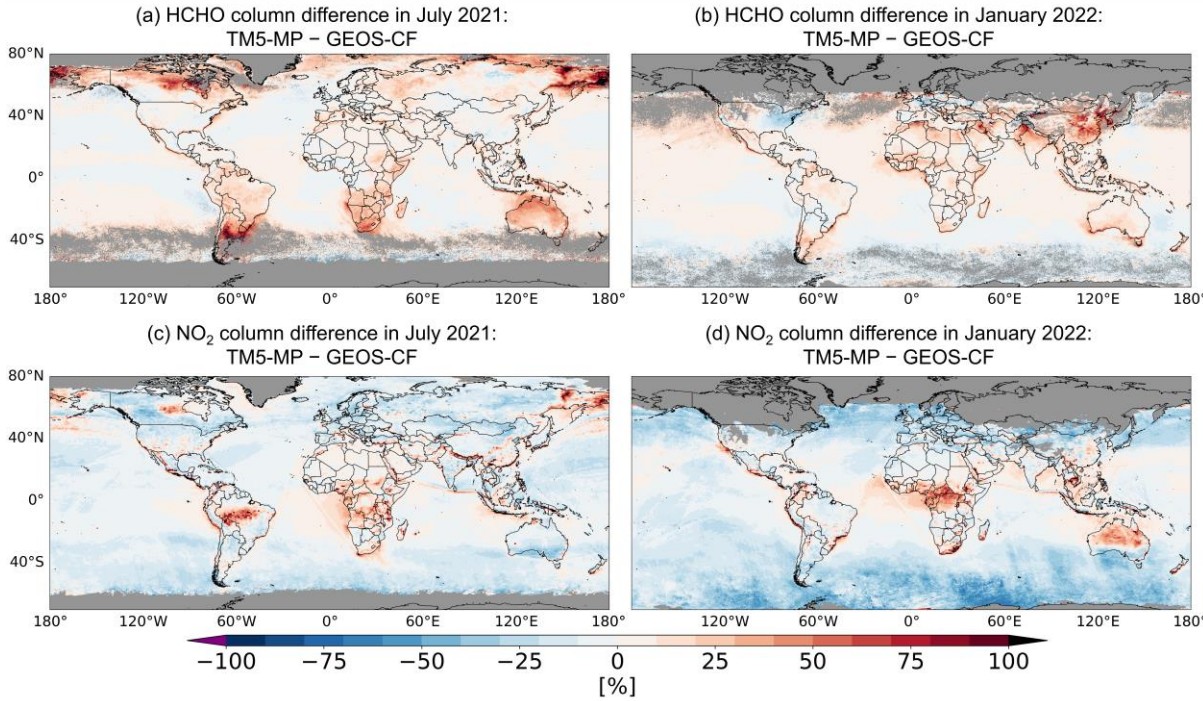


**Figure 9.** Relative differences of tropospheric HCHO (**a** and **b**) and $NO_2$ (**c** and **d**) columns retrieved with TM5-MP priori
profiles (Cases "Fst_tm5" and "Nst_tm5") to those with GEOS-CF priori profiles (Case "Fst_ORcp" and POMINO $NO_2$) in
July 2021 and January 2022. The regions in gray mean that there are no valid observations.
**4.5 Summarizing the impacts of input parameters**
As shown in each sub-figure of Figure 10, the first three columns summarize the structural uncertainty of aerosol
correction, surface reflectance and a priori profile shapes on the HCHO retrieval in the corresponding region and
month. As noted in Sect. 2.2, we consistently use GEOS-CF HCHO columns for background correction in every
HCHO sensitivity test case. The TM5-MP HCHO columns over background regions are systematically lower than
those of GEOS-CF by about $0.5 \times 10^{15}$ molec.cm$^{-2}$ on average (Fig. S11), which strongly affects the comparisons
over the low-HCHO regions.
Over clean areas (HCHO columns < $5 \times 10^{15}$ molec.cm$^{-2}$), a priori profile shapes are the primary source of the
HCHO structural uncertainty (third column in Fig. 10). However, the differences between "Fst_tm5" and the
reference case "Fst_ORcp" are not in alignment with those of RPRO to the reference case, as manifested in the
consistent drop of the blue line from the third ("Fst_tm5" – reference) to the fourth column (RPRO − reference).
This drop can be attributed to the systematic issue in the background correction. Over most areas with HCHO
columns larger than $5 \times 10^{15}$ molec.cm$^{-2}$, relative to the same reference case, the HCHO differences caused by
using implicit aerosol corrections and TM5-MP priori profile shapes match well with those of RPRO product (the
fourth column). However, the lower values of RPRO than the reference case in Europe in January 2022 do not
agree with the combined results of tests "Fst_imaer" and "Fst_tm5". This indicates that the higher OMI-based
climatology monthly MLER used in RPRO retrieval is probably the dominant factor. Furthermore, the influence
of cloud correction using different cloud parameters, especially the cloud top pressures, varies from −20% to 20%
depending on the specific regions and seasons. This is also an important factor for the HCHO differences between
POMINO and RPRO retrievals.

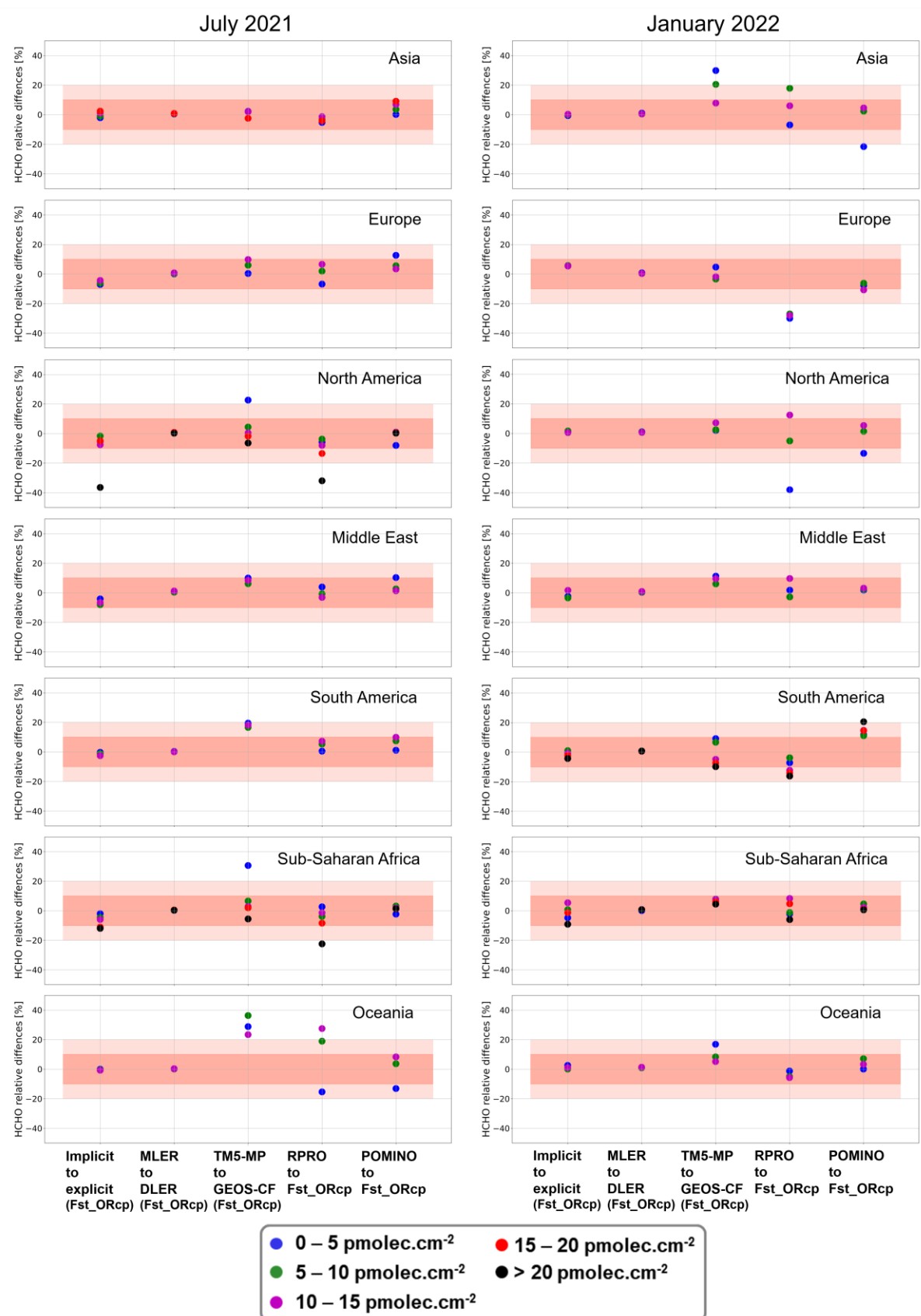

**Figure 10.** HCHO relative differences of the sensitivity test "Fst_imaer" (Case F2, first column), "Fst_mler" (Case F3, second column), "Fst_tm5" (Case F4, third column), RPRO product (fourth column) and POMINO product (fifth column) to the reference "Fst_ORcp" (Case F1) over seven regions in July 2021 and January 2022.


For NO$_2$, the first three columns in Figure 11 show the individual effect of each input parameter on the NO$_2$
retrieval in each region. Apparently, the relative differences between RPRO and POMINO (the fifth column) are
in discrepancy with the sum of the differences between each of the three cases ("Nst_imaer", "Nst_dler" and
"Nst_tm5") and the reference POMINO retrieval, especially over polluted areas in North America, Europe and
Asia in January 2022. However, the NO$_2$ columns of the test "Nst_joint" (Case N4) show high agreement with
those of the RPRO product when compared to the POMINO retrieval (fourth column in Fig. 11); a similar result
is shown for the spatial distribution in Figure S13. Therefore, the NO$_2$ differences between POMINO and RPRO
are the result of compensation effects between different aerosol corrections on one hand, and different surface
reflectances as well as vertical profile shapes on the other hand. These results demonstrate the non-linear joint
effects of aerosols, surface reflectance, clouds and a priori profiles in the AMF calculation, which are consistent
with the previous findings (Lin et al., 2015; Liu et al., 2020). The remaining differences between "Nst_joint" and
RPRO NO$_2$ columns are caused by their different ways to obtain tropospheric NO$_2$ AMFs, i.e., online pixel-
specific RT calculation versus LUT-based interpolation (Lin et al., 2014).

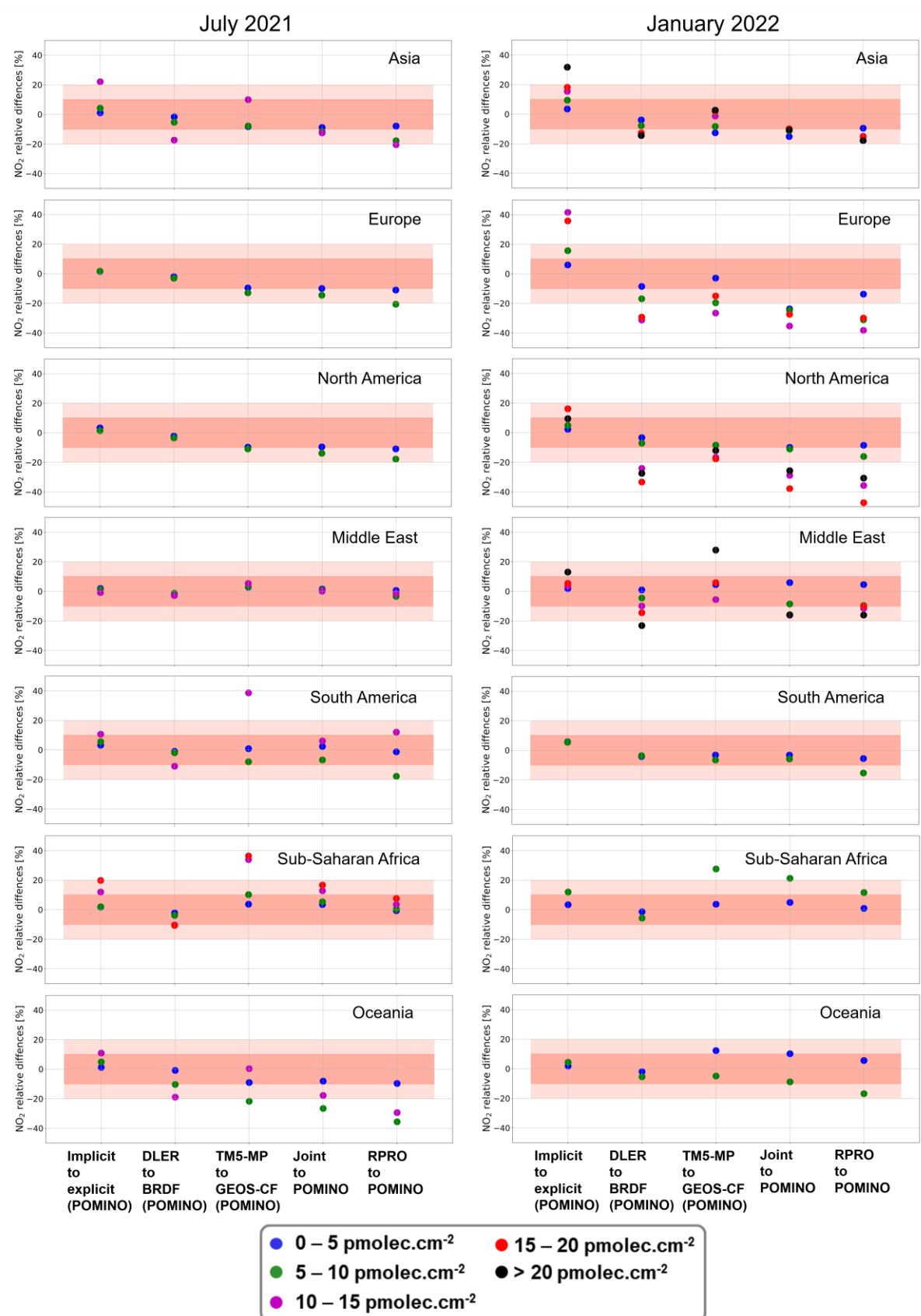


**Figure 11.** NO$_2$ relative differences of the sensitivity test "Nst_imaer" (Case N1, first column), "Nst_dler" (Case N2, second column), "Nst_tm5" (Case N3, third column), "Nst_joint" (Case N4, fourth column) and RPRO product (fifth column) to POMINO product as the reference (Case N0) over seven regions in July 2021 and January 2022.

## 5 Uncertainty estimates

The theoretical uncertainties of the POMINO retrievals can be analytically derived by uncertainty propagation based on the Eqs. 2 and 3 (Boersma et al., 2004). However, it is difficult to estimate the overall AMF uncertainty for each pixel, as one challenge is the amount of computational costs of sensitivity calculations with the online pixel-by-pixel RT simulations. Nonetheless, random uncertainties of the observations can be reduced by spatial and temporal averaging, although the systematic uncertainties from the main retrieval steps remain. There remains lack of information to separate random and systematic uncertainties accurately. Here we provide a preliminary estimate of the uncertainty budget for monthly averaged HCHO and $NO_2$ columns from POMINO retrievals (Tables 3 and 4), based on our sensitivity tests and validations as well as previous work.

For HCHO, the systematic differential slant column uncertainty is 25% for regions with low columns and 15% for regions with elevated columns (De Smedt, 2022; De Smedt et al., 2018). The background correction uncertainty is significant for low columns (around 40%), in which the systematic uncertainty from the dSCD normalization is estimated to be 0 to $4 \times 10^{15}$ molec.cm$^{-2}$, and the uncertainty from the model background is 0 to $2 \times 10^{15}$ molec.cm$^{-2}$. The AMF uncertainty, which is the largest contributor to the vertical column uncertainty, is mainly dependent on the errors of the ancillary parameters tested in Sect. 4. The AMF uncertainty induced by the error of a priori profile shapes is the largest with 30% to 60% over clean regions and around 20% over polluted regions. The errors of cloud parameters and surface reflectance are assumed to contribute to the AMF uncertainty by 10% to 20%, and the errors in the aerosol parameters contribute to the AMF uncertainty by about 5% for regions with low columns and 10% for regions with elevated columns. In addition, the uncertainty due to the aerosol overcorrection issue for partly cloudy pixels with low cloud height is estimated 10% to 15% (Sect. 4.1.1). Overall, the HCHO AMF uncertainty is estimated to be about 70% for clean regions and 30% for polluted regions, respectively.

**Table 3.** Estimated uncertainty budget of POMINO HCHO vertical columns for monthly mean low and elevated columns (higher than $10 \times 10^{15}$ molec.cm$^{-2}$).

| | Remote regions / low columns | Elevated column regions / periods |
|---|:---:|:---:|
| Differential slant column uncertainties (De Smedt, 2022) | 25% | 15% |
| Background correction uncertainties (De Smedt, 2022) | 40% | 10% |
| •   dSCD normalization uncertainties | $0 - 4 \times 10^{15}$ molec.cm$^{-2}$ | |
| •   model background uncertainties | $0 - 2 \times 10^{15}$ molec.cm$^{-2}$ | |
| AMF uncertainties | 70% | 30% |
| •   from a priori profiles uncertainties | 60% | 20% |
| •   from aerosol correction uncertainties | 5% | 10% |
| •   from surface reflectance uncertainties | 20% | 10% |
| •   from cloud correction uncertainties | 20% | 10% |
| •   from aerosol overcorrection issue uncertainties (only for partly cloudy pixels with low clouds) | 15% | 10% |
| Tropospheric vertical column uncertainty | 85% | 35% |

For $NO_2$, the total SCD uncertainty is reported to be 0.5 to $0.6 \times 10^{15}$ molec.cm$^{-2}$ and a constant value of $0.2 \times 10^{15}$ molec.cm$^{-2}$ is assigned to the uncertainty of the stratospheric SCDs (Van Geffen et al., 2022b). For tropospheric AMF, the uncertainty caused by aerosol-related errors is estimated to be 10% to 20%, and the errors in a priori $NO_2$ profile shapes is estimated to cause an AMF uncertainty of 10% on average based on the sensitivity test. The contribution from cloud parameters and surface reflectance to the $NO_2$ AMF uncertainty is estimated to

be on the same level as that of a priori profile shapes. For pixels partly covered by low clouds over both clean and polluted regions, the AMF uncertainty contributed from the aerosol overcorrection issue is within 10%. By adding these errors in quadrature, the overall $NO_2$ AMF uncertainty is 25% to 30%.

**Table 4.** Estimated uncertainty budget of monthly mean POMINO $NO_2$ vertical columns.

| | All regions |
|---|---|
| Total slant column uncertainties (Van Geffen et al., 2022b) | $0.5 - 0.6 \times 10^{15}$ molec.cm$^{-2}$ |
| Stratospheric slant column uncertainties (Van Geffen et al., 2022b) | $0.2 \times 10^{15}$ molec.cm$^{-2}$ |
| AMF uncertainties | 25% – 30% |
| • from a priori profiles uncertainties | 10% |
| • from aerosol correction uncertainties | 10% – 20% |
| • from surface reflectance uncertainties | 10% |
| • from cloud correction uncertainties | 10% |
| • from aerosol overcorrection issue uncertainties (only for partly cloudy pixels with low clouds) | 10% |
| Tropospheric vertical column uncertainty | $0.3 \times 10^{15}$ molec.cm$^{-2}$ + [0.2 to 0.4] × VCD |

Note: the uncertainty in the total slant columns is mostly absorbed by the stratosphere-troposphere separation step, and may not propagate into the tropospheric slant columns. (Van Geffen et al., 2015)

By wrapping up the estimated relative contributions to the vertical column uncertainty, the total uncertainty of POMINO HCHO VCDs is estimated to be 85% over regions with low columns, and 35% over regions with high columns. For the POMINO $NO_2$ retrieval, the overall uncertainty budget can be approximated as $0.3 \times 10^{15}$ molec.cm$^{-2}$ + [0.2 to 0.4] × VCD. This tentative estimation of the POMINO retrieval uncertainties is in agreement with the error analysis by De Smedt (2022) and Van Geffen et al. (2022b), and is supported by the validation results against the independent ground-based measurements (Sect. 6.1). Quantification of the errors at an individual pixel level have been achieved in previous studies (Boersma et al., 2004; Chong et al., 2024; Van Geffen et al., 2022b). As an alternative option to the Gaussian error propagation method, artificial-intelligence-based methods are an appealing approach to be tried in our future work.

**6 Validation against global MAX-DOAS network and PGN measurements**

In this section, we present the validation results of POMINO and RPRO retrievals against independent ground-based measurements from the global MAX-DOAS network and PGN. Separate comparisons of tropospheric HCHO and $NO_2$ columns are given in Sect. 6.1, the effect of vertical smoothing is discussed in Sect. 6.2, and the satellite-based and ground-based FNRs are evaluated in Sect. 6.3.

**6.1 Validation of tropospheric HCHO and $NO_2$ columns**

Figures 12a and b present the scatterplots of daily satellite HCHO columns against ground-based measurements in April, July, October 2021 and January 2022. Each data point represents a day and site. There is a lower slope and higher positive offset for POMINO compared with those of RPRO product (slope: 0.56 versus 0.61; offset: 1.17 versus 0.24). This is in line with the discussion in Sect. 4.5 that POMINO employs higher HCHO columns from GEOS-CF for background correction, which is the major component of HCHO columns over areas with low HCHO level. Furthermore, at 13 polluted ground-based sites where HCHO columns are higher than $10 \times 10^{15}$ molec.cm$^{-2}$, POMINO HCHO columns show smaller bias at 8 sites (Figure S14). Overall, POMINO exhibits a

smaller negative NMB (−30.8%) than RPRO (−35.0%). Statistics of separate validation results against MAX-
DOAS and PGN measurements are given in Table S3.

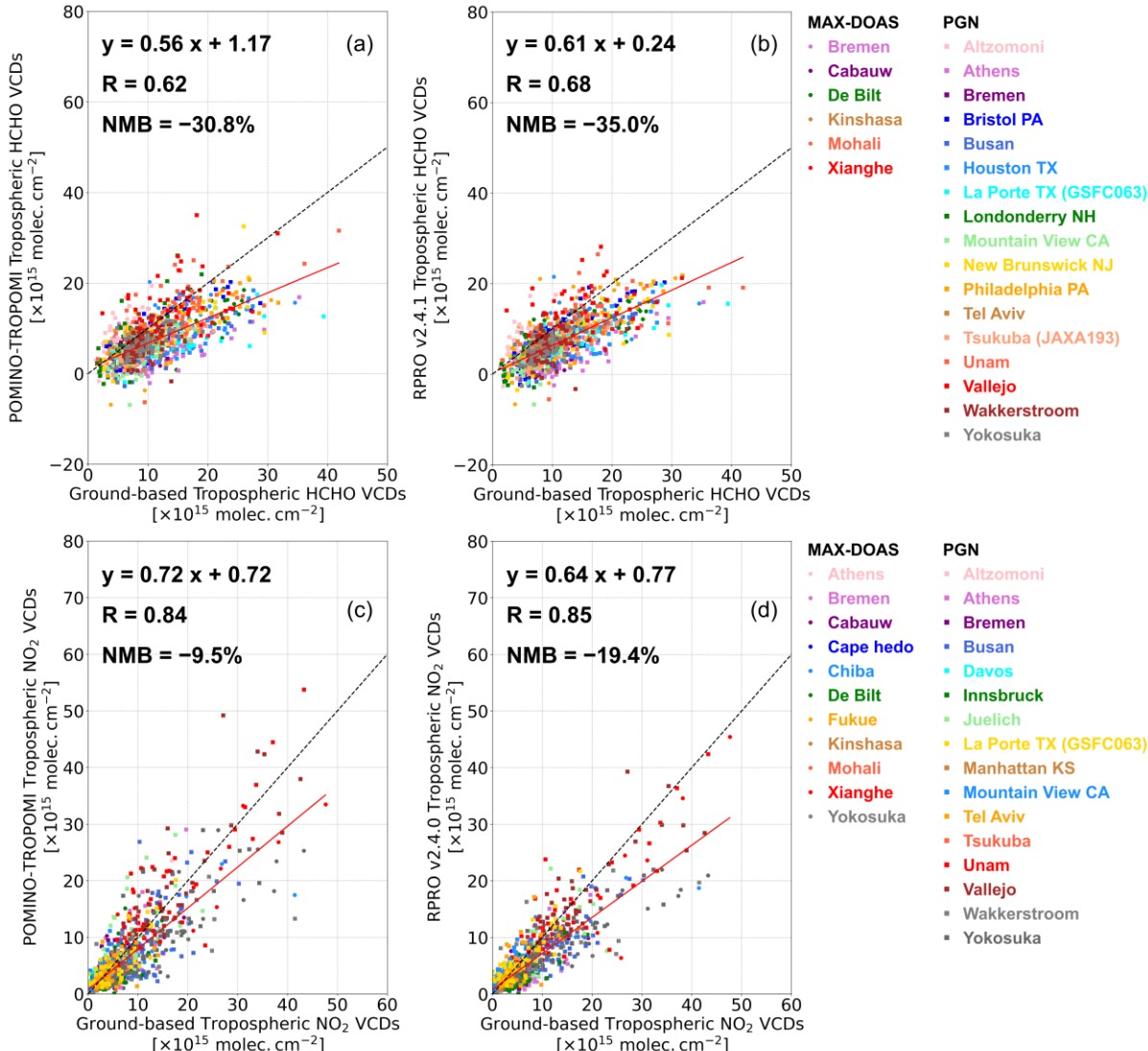


**Figure 12.** Scatterplots of tropospheric HCHO (**a** and **b**) and NO$_2$ (**c** and **d**) columns between satellite products (POMINO and
RPRO) and ground-based measurements in April, July, October 2021 and January 2022. The slope, offset and correlation from
a linear regression using the robust Theil-Sen estimator and normalized mean bias (NMB) are given in each panel and plotted
as the red line. The black dashed line is the 1:1 line. Each MAX-DOAS (marked by circles) and PGN site (marked by squares)
is color-coded and listed on the right side.
For NO$_2$, a better agreement with ground-based measurements is found for POMINO tropospheric columns than
for RPRO (slope: 0.72 versus 0.64; offset: 0.72 versus 0.77; NMB: −9.5% versus −19.4%). At remote MAX-
DOAS sites where tropospheric NO$_2$ columns are around $1 \times 10^{15}$ molec.cm$^{-2}$ or less (Fig. S14), satellite
tropospheric NO$_2$ columns are higher by 0.3-1 $\times 10^{15}$ molec.cm$^{-2}$. This is in line with the previous validation
studies (Kanaya et al., 2014; Pinardi et al., 2020; Verhoelst et al., 2021; Zhang et al., 2023), and is probably
because that a majority of NO$_2$ molecules over remote regions are in the free troposphere, which are above the
detection height of ground-based MAX-DOAS instruments but can be well observed by spaceborne instruments.
At the six most-polluted sites with mean tropospheric NO$_2$ columns higher than $10 \times 10^{15}$ molec.cm$^{-2}$, POMINO
features a much-reduced bias of −14.5% compared with RPRO product (−22.0%). This is because of the explicit

correction for aerosol "shielding" effect over highly polluted sites and lower surface reflectance, which reduces the $NO_2$ scattering weights near the surface and hence increases the retrieved $NO_2$ columns.

## 6.2 Effect of vertical smoothing for validation

To test the impact of different vertical sensitivity from the ground and space, MAX-DOAS FRM4DOAS v01.01 harmonized HCHO and $NO_2$ datasets were used. The data provides 20-layer-resolved (from surface to ~ 600 hPa) MAX-DOAS averaging kernels and vertical profiles (posterior and prior to the retrievals). Following the "vertical smoothing" technique (Rodgers and Connor, 2003) described in detail by Vigouroux et al. (2020), we first substituted the priori profile shapes used in MAX-DOAS retrieval with either GEOS-CF or TM5-MP profile shapes to get corrected MAX-DOAS retrieved profiles:

$$x'_{\text{MD}} = x_{\text{MD}} + (\mathbf{A}_{\text{MD}} - \mathbf{I})(x_{\text{MD,a}} - x_{\text{Sat,a}}) \tag{5}$$

with $x'_{\text{MD}}$ denoting the corrected MAX-DOAS retrieved profile, $x_{\text{MD}}$ the original MAX-DOAS profile, $\mathbf{A}_{\text{MD}}$ the MAX-DOAS averaging kernel matrix, $\mathbf{I}$ the unit matrix, $x_{\text{MD,a}}$ the MAX-DOAS a priori profile and $x_{\text{Sat,a}}$ the satellite a priori profile (i.e., from GEOS-CF or TM5-MP) re-gridded to the MAX-DOAS retrieval resolution from the surface to 600 hPa. To account for the trace gas content in the free troposphere, especially for HCHO, we further extend the corrected MAX-DOAS profile to the tropopause with the satellite profile above 600 hPa that is scaled to ensure vertical continuity of the overall tropospheric profile. After that, we perform the smoothing process using either POMINO or RPRO averaging kernels:

$$c_{\text{MD}}^{\text{smoothed}} = a_{\text{Sat}} \cdot x'_{\text{MD}} \tag{6}$$

with $c_{\text{MD}}^{\text{smoothed}}$ the smoothed MAX-DOAS column, $a_{\text{Sat}}$ the satellite averaging kernel vector and $x'_{\text{MD}}$ the corrected MAX-DOAS retrieved profile from Eq. (5). We compare the smoothed MAX-DOAS data with satellite retrievals and the statistics are summarized in Table 5.

For the five MAX-DOAS sites available (Table 2), we find that after smoothing, the linear regression slope gets improved for both HCHO products. The negative bias of POMINO is reduced by about 10% but that of RPRO product is increased by about 4%. This is because POMINO HCHO averaging kernels are smaller than those of RPRO between the surface to about 800 hPa, resulting in lower smoothed MAX-DOAS HCHO columns compared to those using RPRO HCHO averaging kernels. Smaller POMINO HCHO averaging kernels at low altitudes are due to enhanced "shielding" effect from explicit aerosol corrections and lower KNMI TROPOMI MLER than OMI-based climatological monthly MLER used in RPRO HCHO.

For $NO_2$, among the six sites (Table 2), after applying the vertical smoothing technique, the negative NMB increases from −7.3% to −15.7% for POMINO and decreases from −24.6% to −8.5% for RPRO, even though a better day-to-day correlation is found for both products. Again, such changes are caused by the different averaging kernels used in the two satellite products.

Due to the scarcity of the MAX-DOAS sites for analysis here (Tables 2 and 5) and the under-representativeness in their spatial distribution (Table 2), a general conclusion cannot be made on the overall impact of vertical smoothing now. Nevertheless, the comparison results indicate the importance of considering the different vertical sensitivity between spaceborne and ground-based MAX-DOAS instruments, and different a priori profile shapes used to derive the vertical columns during the validation practice (De Smedt et al., 2021; Dimitropoulou et al., 2022; Yombo Phaka et al., 2023).

**Table 5.** Effect of vertical smoothing on the comparisons of TROPOMI and MAX-DOAS data.

| HCHO (five sites) | Direct comparisons | | Vertical smoothing applied | |
|---|---|---|---|---|
| | POMINO | RPRO | POMINO | RPRO |
| Slope | 0.56 | 0.65 | 1.08 | 0.72 |
| Offset [$10^{15}$ molec.cm$^{-2}$] | 2.15 | 0.18 | −1.58 | −0.78 |
| Correlation | 0.63 | 0.66 | 0.66 | 0.73 |
| NMB | −22.6% | −30.8% | −10.9% | −34.2% |

| NO$_2$ (six sites) | Direct comparisons | | Vertical smoothing applied | |
|---|---|---|---|---|
| | POMINO | RPRO | POMINO | RPRO |
| Slope | 0.80 | 0.64 | 0.72 | 0.74 |
| Offset [$10^{15}$ molec.cm$^{-2}$] | 0.38 | 0.46 | 0.74 | 0.98 |
| Correlation | 0.81 | 0.84 | 0.90 | 0.86 |
| NMB | −7.3% | −24.6% | −15.7% | −8.5% |

## 6.3 Comparisons of FNR

The FNR is an important space-based indicator of the ozone chemistry regimes and its sensitivity to precursor emissions. Figure 13 shows the scatterplots of daily FNR derived from POMINO and RPRO products against ground-based measurements, i.e., MAX-DOAS and PGN, in April, July, October 2021 and January 2022. A better agreement is found between POMINO and ground-based FNR with improved linear regression statistics (slope: 0.73 versus 0.69; offset: 0.15 versus 0.22) and reduced NMB (−14.8% versus −21.1%) compared to those of RPRO products. Moreover, the regression results are better in the comparisons for FNR than those in the individual comparisons for either HCHO or NO$_2$ tropospheric VCDs (Sect. 6.1). This demonstrates the potential of using POMINO HCHO and NO$_2$ retrievals to improve the studies on the ozone sensitivity analysis for NO$_x$ as well as VOC emission controls.

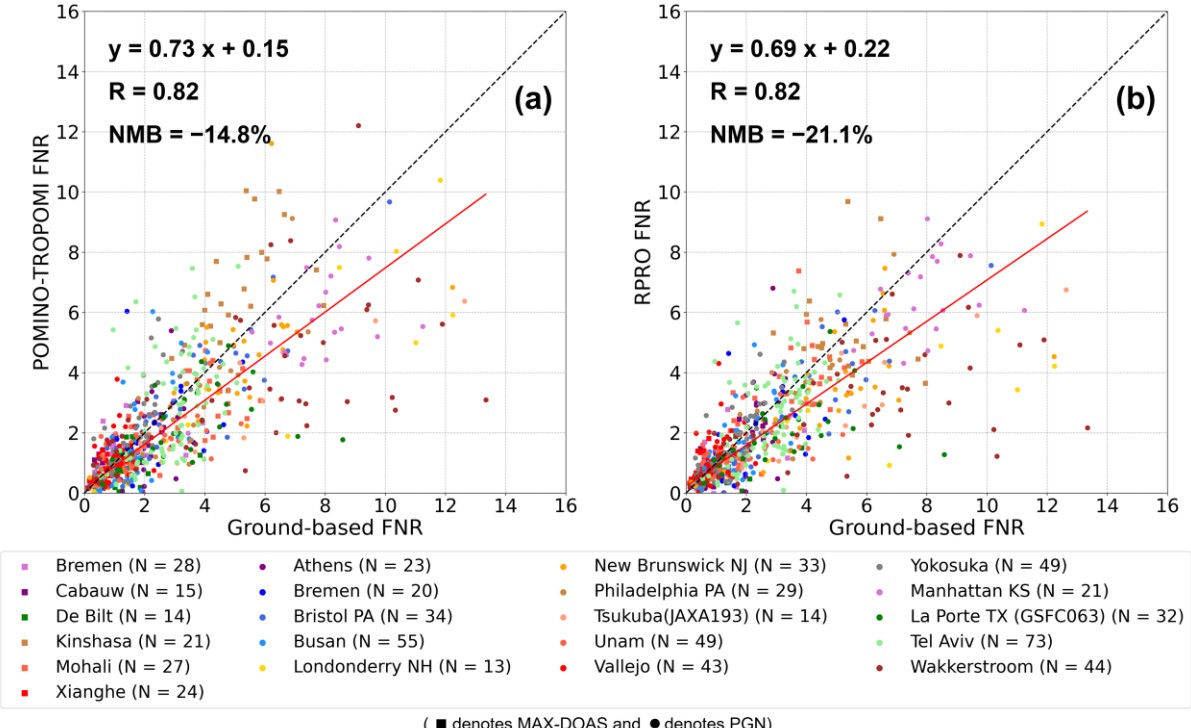

**Figure 13.** Scatterplots of daily tropospheric column ratio of formaldehyde to nitrogen dioxide (FNR) derived from satellite products (**a** for POMINO and **b** for RPRO) and ground-based measurements in April, July, October 2021 and January 2022. The slope, offset and correlation from a linear regression using the robust Theil-Sen estimator and normalized mean bias (NMB) are given in each panel and plotted as the red line.

Note that most ground-based sites used here are in the North America, Europe, South Korea and Japan, but very
few or even no sites in other countries or continents (Figure S2). Thus, further validation with ground-based
measurements in combination with model simulations is needed over other regions, especially those where ozone
chemistry regimes change rapidly.
**7 Conclusions**
We developed an updated version of the POMINO algorithm providing HCHO and $NO_2$ AMF calculations, which
offers global tropospheric HCHO and $NO_2$ VCDs retrievals of TROPOMI with improved consistency compared
to current products. Compared to the independently developed RPRO HCHO and $NO_2$ operational algorithms
using different ancillary parameters, the POMINO algorithm includes: (1) the surface reflectance anisotropy by
using KNMI TROPOMI v2.0 DLER at 340 nm for HCHO and MODIS BRDF coefficients around 470 nm for
$NO_2$, (2) an explicit aerosol correction for both species based on GEOS-CF aerosol information and MODIS AOD
at corresponding wavelengths, (3) high-resolution (0.25º × 0.25º) a priori HCHO and $NO_2$ profile shapes from
GEOS-CF dataset and (4) a consistent cloud correction based on cloud top pressures taken from the FRESCO-S
cloud product and cloud fractions re-calculated at 440 nm using the same ancillary parameters as those used in
$NO_2$ AMF calculation.
High qualitative agreement of tropospheric HCHO and $NO_2$ columns is found between POMINO and RPRO
products in April, July, October 2021 and January 2022. However, RPRO HCHO columns are lower by 15% on
average than the POMINO HCHO columns over the polluted areas around the world, and the negative differences
of RPRO tropospheric $NO_2$ columns can reach −20% over specific areas.
To clarify the reasons for the differences between POMINO and RPRO columns and quantify the structural
uncertainty from ancillary parameters in the AMF calculation, we performed a series of sensitivity tests on the
cloud correction, aerosol correction, surface reflectance and a priori profile shapes. We find that based on
POMINO-recalculated cloud fraction at 440 nm and FRESCO-S cloud top pressures, differences between clear-
sky AMFs and total AMFs vary from −25% to more than 50% for both HCHO and $NO_2$, depending on the cloud
fraction and the relative height between clouds and trace gases. When using cloud top pressure data from
OCRA/ROCINN-CRB instead of FRESCO-S, a large decrease of tropospheric HCHO columns is found (> 2 ×
$10^{15}$ molec.cm$^{-2}$) over Amazonia Rainforest and southeast China, and the negative differences over polluted
regions are about 20% on average.
The influence of the implicit aerosol corrections used in operational products is within 10% on the HCHO retrieval,
while higher $NO_2$ columns by 20% to 40% over the polluted areas in January 2022 are found with implicit aerosol
corrections. Comparisons of retrieved $NO_2$ columns using clear-sky AMFs and total AMFs with implicit aerosol
corrections prove that the positive difference for $NO_2$ is dominated by the enhanced "shielding" effect of clouds
over $NO_2$ layers. The directionality of the surface reflectance has a very small impact on the HCHO retrieval in
the UV band, but the structural uncertainty of surface reflectance for $NO_2$ over polluted areas can reach 30%. The
HCHO structural uncertainty from a priori profile shapes is 20% to 30% over the background areas and 10% to
20% over the polluted areas. In contrast, the $NO_2$ differences due to different a priori profile shapes reach 30% or
more over the polluted areas. The additional test on the joint effect of these parameters shows notable non-linear
influences from aerosol correction, surface reflectance, cloud correction and a priori profile shapes in the RT
calculation.

Direct comparisons of tropospheric HCHO and $NO_2$ columns between satellite retrievals and ground-based measurements from the global MAX-DOAS network and PGN show that both POMINO HCHO and $NO_2$ retrievals feature a reduced bias in comparison to RPRO products (HCHO: −30.8% versus −35.0%; $NO_2$: −9.5% versus −19.4%), especially at the polluted sites. The effect of the vertical smoothing is significant and strongly depends on the satellite averaging kernels. A better agreement of daily FNR with smaller bias is also found between POMINO products and PGN measurements in comparison to results obtained with RPRO products (NMB: −14.8% versus −21.1%).

Overall, we demonstrate the promising performance of TROPOMI-based POMINO algorithm for global HCHO and $NO_2$ retrieval. However, there are still several limitations in our study. First, the aerosol overcorrection issue for partly cloudy pixels exists in the current POMINO algorithm, which has been discussed in detail in Sect. 4.1.1. The uncertainty due to this issue is estimated to be within 15% for HCHO and 10% for $NO_2$. Given that TROPOMI-based $O_2$-$O_2$ cloud data have become available, we plan to improve the current POMINO algorithm by performing $O_2$-$O_2$ cloud retrieval for both cloud fraction and cloud top pressure with explicit aerosol corrections in the future, as has been done in the POMINO-OMI and POMINO-GEMS products (Lin et al., 2015; Liu et al., 2019; Zhang et al., 2023).

Second, it should be noted that the indirect aerosol effect on HCHO and $NO_2$ retrievals through clouds is strongly sensitive to the cloud top pressures and the trace gas profile shapes. Using OMI $O_2$-$O_2$ based cloud parameters or FRESCO-S cloud top pressures stored in the operational $NO_2$ L2 product before version 1.4.0, previous studies have shown lower $NO_2$ columns over polluted North China Plain when retrieved with implicit aerosol corrections (Lin et al., 2015; Liu et al., 2020). This is because the cloud top pressures in those studies are higher, which result in larger AMF values when implicit (instead of explicit) aerosol corrections are used. Besides, certain biases still exist in the current FRESCO-S cloud top pressures, such as the overestimation over the ITCZ. The effect of a priori profile shapes is also significant for both HCHO and $NO_2$ retrievals, and it deserves more attention in the future analysis. Comprehensive evaluations of cloud retrievals and model performance with independent measurements are needed in future studies.

Nevertheless, the POMINO algorithm that aims at improving the consistency in multi-gas retrieval shows great potential and can be easily adapted to other satellite instruments, e.g. GEMS, TEMPO, as well as Sentinel-4 and Sentinel-5 missions. The global tropospheric HCHO and $NO_2$ VCD retrievals presented in our study are also of value for subsequent applications such as ozone chemistry analysis and emission controls.

*Data availability.* The POMINO HCHO and $NO_2$ datasets will be available soon on our website (http://www.pku-atmos-acm.org/acmProduct.php/). Before release, the data presented in the study are available from the corresponding authors upon request. The S5p TROPOMI RPRO HCHO v2.4.1 L2 product and RPRO $NO_2$ v2.4.0 L2 product are available at Copernicus Data Space Ecosystem | Europe's eyes on Earth (https://dataspace.copernicus.eu/, last access: 17 July 2024). The ground-based MAX-DOAS measurements can be provided upon request to the corresponding authors. The PGN/Pandora direct sun measurements are available at the ESA Validation Data Centre (EVDC, 2024) (https://evdc.esa.int, last access: 7 July 2024) and Pandonia Global Network (2024) (https://www.pandonia-global-network.org/, last access: 17 July 2024).

*Supplement.*

848

*Author contributions.* YZ, JL, NT and MVR conceived this research. YZ, HY, IDS, JL, NT and MVR designed the algorithm. YZ, HY, IDS, JL, MVR, GP, AM and SC designed the validation process together. YZ performed all calculations. RS provided LIDORT model. RN, FR, SW, LC, JVG, ML, WS and LF provided data and technical support for satellite retrievals. GP and SC provided methodological support for validation. AMC is the network principal investigator (PI) for PGN instruments. GP, SC, AMC and MT provided the discussion for PGN uncertainty estimation. MVR, GP, AM, MMF, AR, AP, VK, VS, TW, YC, HT, YK and HI provided ground-based MAX-DOAS measurements. YZ wrote the paper with inputs from JL, NT, IDS and MVR. All co-authors revised and commented on the paper.

*Competing interests.* At least one of the (co-)authors is a member of the editorial board of Atmospheric Measurement Techniques. The authors have no other competing interests to declare.

*Acknowledgements.* This work contains modified S5p TROPOMI L1b data post-processed by BIRA-IASB. We thank all the colleagues for supporting the ground-based MAX-DOAS measurements used in this work. The PGN is a bilateral project supported with funding from NASA and ESA. We thank all the PIs, support staff and funding for establishing and maintaining all the PGN sites used in this investigation.

*Financial support.* This work has been supported by the National Natural Science Foundation of China (grant no. 42075175 and 42430603), the National Key Research and Development Program of China (grant no. 2023YFC3705802), the China Scholarship Council (CSC) (grant no. 202306010348), and the Royal Belgian Institute for Space Aeronomy (BIRA-IASB) (project code: 3TEAMUVVIS).

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
