# Peer review of "Global retrieval of TROPOMI tropospheric HCHO and NO2 1"

_Atmospheric Measurement Techniques, 2024_

## Author Response (AR1)

**Authors' response to comments from Anonymous Referee #1**

**General comments:**

The paper "Global retrieval of TROPOMI tropospheric HCHO and $NO_2$ columns with improved consistency based on updated Peking University OMI $NO_2$ algorithm", by Zhang and co-authors, presents an updated version of the POMINO algorithm for HCHO and $NO_2$ and its structural uncertainty in the AMF calculation. The paper is well structured. The topic fits in the scope of AMT.

We sincerely thank the Referee #1 for reviewing our paper and providing constructive comments for improvement. Responses to the comments are provided below.

**Specific comments:**

One concern is the application of aerosol correction on FRESCO parameters. The FRESCO CF has been recalculated and corrected using aerosol information in UV and VIS, but the original $O_2$-A band may be less affected by aerosol than UV/VIS, possibly introducing an overcorrecting issue. Such overcorrecting can be amplified by using the un-corrected CP.

Thank you very much for this comment. A similar comment was raised by Referee #3.
We agree that the aerosol overcorrection issue occurs for partly cloudy pixels because we only recalculate the cloud fraction with explicit aerosol corrections, but use the cloud top pressure from FRESCO-S product which already implicitly includes aerosols. Liu et al. (2020) quantified such overcorrection issue for aerosols by conducting a sensitivity study using a "semi-explicit" aerosol correction approach. In this approach, aerosol optical effects are explicitly corrected for clear-sky AMFs, but are excluded for the cloudy-sky portion of partly cloudy pixels. Results show that $NO_2$ differences due to the aerosol correction choice for cloudy-sky AMFs vary from 3.1% to 11.2% over East Asia in July 2018, depending on the $NO_2$ pollution level (Section 3.3 in Liu et al., 2020). It should be noted that the FRESCO-S cloud top pressure data stored in the v1.2–v1.3 $NO_2$ data product, as used by Liu et al. (2020), are reported with a high bias over scenes with low cloud fractions and/or a considerable aerosol load. An improved version based on the FRESCO-wide approach is applied in v1.4 and subsequent $NO_2$ products, which was proven to be more realistic compared with the old version (Van Geffen et al., 2022a, b).
Using the updated FRESCO-S cloud top pressure data stored in the RPRO v2.4.0 $NO_2$ product, we attempt to estimate the impact of the aerosol overcorrection issue without conducting a sensitivity study. Given the fact that, in the retrieval algorithm, the cloud is assumed to be an optically thick Lambertian reflector with a high albedo of 0.8, it is reasonable to assume that the impact of this overcorrection becomes non-negligible when the FRESCO-S cloud top pressure is too high, meaning the cloud is very close to the surface and therefore vertically mixed with aerosols and trace gases; while for pixels where the cloud is higher than the trace gases, the aerosol correction should have little influence on the cloudy-sky AMFs. Based on this strategy, we selected all valid pixels where the difference between the surface pressure and the FRESCO-S cloud top pressure is equal to 100 hPa or less in July 2021 and January 2022. In such case, we can mitigate the potential aerosol overcorrection by using aerosol-corrected clear-sky AMFs instead of aerosol-corrected total AMFs.
The comparison result in Figure S6 (shown below) shows that the normalized mean bias (NMB) is around

14% on average for HCHO retrievals (~16% for clean pixels with HCHO column $\leq 10 \times 10^{15}$ molec.cm$^{-2}$, and ~8% for polluted pixels with HCHO column $> 10 \times 10^{15}$ molec.cm$^{-2}$), and around 8% on average for NO$_2$ retrievals. The NO$_2$ results are also qualitatively in line with those in Liu et al. (2020). Therefore, we tentatively estimate the uncertainty due to the aerosol overcorrection to be in the range from 10% to 15% for HCHO and 10% for NO$_2$.

In line 403-429 in the revised manuscript, we added:

"One issue existing in the process of cloud correction in the POMINO retrieval is that only the cloud fraction is re-calculated with explicit aerosol corrections, while the cloud top pressure is taken from the external dataset, i.e., the FRESCO-S cloud product, in which the aerosols are implicitly accounted for. As a result, this step introduces presumably an aerosol overcorrection issue in the cloud top pressures of partly cloudy pixels, and therefore brings in additional uncertainties in the AMF calculations. Lin et al. (2015) reported that excluding aerosols leads to an increase of O$_2$-O$_2$-based cloud top pressures (from 700–900 hPa to 750–950 hPa) over eastern China, but it is difficult to clarify the mechanism due to its complexity (Lin et al., 2014). Currently there is no direct way to estimate the effect of aerosol correction on the FRESCO-S cloud height retrieval without doing O$_2$ A-band cloud retrieval tests, which is beyond the scope of this study. However, below we give an estimation of the uncertainty in POMINO HCHO and NO$_2$ vertical columns caused by this issue.

Given the fact that, in the retrieval algorithm, the cloud is assumed to be an optically thick Lambertian reflector with a high albedo of 0.8, the cloudy-sky AMF (and hence tropospheric AMF) is very sensitive to the accuracy of the cloud height when the cloud is low and vertically mixed with the aerosols and trace gases. In these cases, we can assume that the retrieved cloud height is primarily influenced by aerosols (Van Geffen et al., 2022a), therefore the aerosol overcorrection issue becomes non-negligible. Focusing on valid pixels for which the difference between the surface pressure and the FRESCO-S cloud top pressure is equal to 100 hPa or less (~17.5% and ~19.9% of total pixels in July 2021 and January 2022, respectively), the aerosol overcorrection uncertainty can be roughly estimated from the difference of HCHO and NO$_2$ vertical columns retrieved using either aerosol-corrected clear-sky AMFs (aerosol correction applied; cloud correction not applied) or aerosol-corrected total AMFs (both aerosol and cloud corrections applied). Based on the results shown in Figure S6, we tentatively estimate the uncertainty to be in the range from 10% to 15% for HCHO, and within 10% for NO$_2$. The estimated NO$_2$ uncertainty level is also supported by the sensitivity test results in Liu et al. (2020). They implemented a "semi-explicit" aerosol correction approach, in which aerosol optical effects are explicitly corrected for clear-sky AMFs, but are excluded for the cloudy-sky portion of partly cloudy pixels, and found the NO$_2$ differences due to the aerosol correction choice for cloudy-sky AMFs vary from 3.1% to 11.2% over eastern China in July 2018. The tentatively estimated uncertainty range above is comparable to or less than that from other ancillary parameters (Sect. 5), and only needs to be taken into account for partly cloudy pixels with low clouds."

In the revised Supplement, we added:

[Figure]

**Figure S6.** Scatterplots of POMINO tropospheric HCHO (**a** and **b**) and NO₂ (**c** and **d**) VCDs retrieved using either aerosol-corrected and cloud-corrected total AMF (x-axis) or aerosol-corrected clear-sky AMF (y-axis), from all pixels where the difference between surface pressure and FRESCO-S cloud top pressure is equal to 100 hPa or less. The left column shows the results for July 2021, and the right column for January 2022. The slope, offset and correlation from a linear regression using the robust Theil-Sen estimator and normalized mean bias (NMB) are given in each panel and plotted as the red line. The black dashed line is the 1:1 line.

Table 1 is a bit heavy, and I recommend to remove redundant information, such as "daily" in a priori profiles. One typo in POMINO HCHO CF is 340 instead of 440.

Thank you for the recommendation. We have simplified Table 1 in the revised manuscript.
Regarding the cloud fraction in POMINO HCHO retrievals, we use values re-calculated at 440 nm (in the NO₂ retrieval) instead of re-calculating them at 340 nm, because (1) the cloud fraction derived at 440 nm is expected to be more reliable than at 340 nm due to the larger noise in the UV band; (2) we want to perform fully consistent cloud corrections in both POMINO HCHO and NO₂ retrievals. We have changed the description to "CF and CP: same as POMINO NO₂" in Table 1 to make it more clear in the revised manuscript.

What is the meaning of "VCD from QA4ECV" for Mohali in Table 2?

It means a retrieval strategy where MAX-DOAS vertical columns are calculated using tropospheric AMFs based on climatological profiles of both trace gas and aerosol loads, as developed during the QA4ECV project (http://uvvis.aeronomie.be/groundbased/QA4ECV_MAXDOAS/QA4ECV_MAXDOAS_readme_website.pdf).
These data are more accurate than the simple geometric approximation strategy as used in previous studies (De Smedt et al., 2021). We have updated the description and added the reference in the revised manuscript.

Please enlarge the font size of xaxis label in Figures 10 and 11.

Done.

**Authors' response to comments from Anonymous Referee #2**

**General comments:**

The paper presents TROPOMI formaldehyde and nitrogen dioxide retrievals. This new algorithm uses the operational slant columns distributed by ESA and focuses on improving the "operational" Air Mass Factors (AMF). It is built on the Peking University OMI nitrogen dioxide algorithm (POMINO). In comparison with the "operational" retrievals the products presented here use improved AMFs. The most important differences is the explicit consideration of aerosols in radiative transfer calculations, the derivation of cloud fractions accounting for the presence of aerosols, the consideration of anisotropic surface reflectance and the use of a higher resolution chemical forecast as source of a priori vertical profiles.
These products show good performance, slightly better than TROPOMI operational products, providing important information regarding the influence of aerosols in AMF uncertainties. Given the clarity of the manuscript and the science these new products may enable the publication of this work to be well justified. The manuscript already has good quality. Hopefully the comments provided below will marginally help to improve the clarity and soundness of the paper.

We sincerely thank the Referee #2 for reviewing our paper and providing constructive comments for improvement. Responses to these specific comments are provided below.

**Specific comments:**

Line 63: When listing past sensors with $NO_2$ and HCHO capabilities would the authors consider adding TEMPO and Suomi-NPP/JPSS satellites that carry the OMPS-NM instrument?

Thank you for the suggestion. We have added TEMPO and OMPS-NM instruments in the revised manuscript.

Line 78: Some $NO_2$ strat/trop separation schemes use AMFs during the process. See for example Bucsela et al. (2013) and Geddes et al. (2018)

Thank you for the comment. In the revised manuscript, we have updated the expression to "while for $NO_2$ a stratosphere-troposphere separation is performed in order to obtain tropospheric columns."

Line 79: Please specify these studies relate to TROPOMI

We have specified the studies in the revised manuscript.

In Line 83-89, we added:

"For example, Liu et al. (2021) present an improved tropospheric $NO_2$ retrieval algorithm from TROPOMI measurements over Europe, which employs a new stratosphere-troposphere separation and updated auxiliary parameters, including a more realistic cloud treatment, for AMF calculation. Over East Asia, Liu et al. (2020) release a new TROPOMI product for tropospheric $NO_2$ columns that features explicit aerosol corrections in the AMF calculation, and Su et al. (2020) improve the TROPOMI tropospheric HCHO retrieval by optimizing the spectral fit and using a priori profiles from a higher resolution regional chemistry transport model."

Line 122: Slightly different spatial resolution for TROPOMI $NO_2$ is reported in van Geffen et al. 2022: 7.2 km x 3.6 km and 5.6 x 3.6 km. Would it be possible to clarify the situation?

Thank you very much for this comment. We have had direct communication with Jos van Geffen regarding this issue. As a matter of fact, the "7.0 km × 3.5 km and 5.5 km × 3.5 km" are approximations, and the "7.2 km × 3.6 km and 5.6 km × 3.6 km" are slightly more accurate numbers. However, the European Space Agency (ESA) prefers that we use the approximate numbers in the communication. For example, in the algorithm theoretical basis document (ATBD) for the TROPOMI L01b data processor (https://sentinel.esa.int/documents/247904/2476257/Sentinel-5P-TROPOMI-Level-1B-ATBD), HCHO v2.4.1 data products (https://sentinels.copernicus.eu/documents/247904/2476257/Sentinel-5P-ATBD-HCHO-TROPOMI) and $NO_2$ v2.4.0 data products (https://sentinel.esa.int/documents/247904/2476257/Sentinel-5P-TROPOMI-ATBD-NO2-data-products.pdf), the description of the TROPOMI spatial resolution at nadir is consistently "approximately 7.0 km × 3.5 km and 5.5 km × 3.5 km". Therefore, we decided to keep using the description "7.0 km × 3.5 km and 5.5 km × 3.5 km" in the revised manuscript.

Line 165: Does POMINO directly use MODIS MYD04_L2 or is it only through GEOS-CF assimilation? Figure S1 seems to indicate the second situation.

Thank you for the comment. In POMINO algorithm, we directly use monthly AOD data from MODIS/Aqua Collection 6.1 MYD04_L2 dataset, with spatial and temporal interpolation for missing values, to constrain the GEOS-CF AOD. We have updated Figure S1 in the revised Supplement to make it less misleading.

Line 177: is it MCD43C2 061or 006, there is conflict with figure S1.

Thank you very much for this. The version of the dataset is 6.1 (https://lpdaac.usgs.gov/products/mcd43c2v061/). We have corrected this typo in the revised manuscript.

Line 202: What is the meaning of structural uncertainty? The experiments described here evaluate the sensitivity with respect to different parameter sources but don't consider the uncertainty on those sources

or how they relate to each other.

The structural uncertainty is the uncertainty that arises when different retrieval methodologies are applied to the same data. We completely agree with your comment. The objective of the sensitivity tests here is to quantify the structural uncertainty of tropospheric HCHO and $NO_2$ AMFs caused by using different but equally valid auxiliary parameters (i.e., cloud information, aerosol information, surface reflectance and a priori profiles), and to explain the corresponding differences between POMINO and RPRO data products. In Section 5, we provide a preliminary estimate of the uncertainty from each parameter source and the uncertainty budget for monthly averaged HCHO and $NO_2$ columns from POMINO retrievals.

Line 236: Pandora HCHO retrievals vary in quality drastically depending on station. They can also degrade their accuracy over time. Have the authors made any effort to quality control Pandora datasets? If so, it would be very helpful to include a description of that work. If not, the situation with Pandora HCHO retrievals should be discussed and acknowledged and if possible mitigated.

Thank you for the comment. We are aware of the issue and agree that efforts for quality control must be made with respect to PGN datasets, especially HCHO retrievals. In our work, we only used PGN data which are submitted to EVDC because this PGN subset undergoes a more thorough quality check. Then, we carefully selected PGN measurements with the label "assured high quality" (data quality flag of 0) and "not-assured high quality" (data quality flag of 10) for the validation (https://www.pandonia-global-network.org/wp-content/uploads/2024/11/PGN_DataProducts_Readme_v1-8-9.pdf).
We have elaborated the discussion of PGN retrieval uncertainties and the description of the PGN data selection in the revised manuscript.
In line 271-286, we added:
"Herman et al. (2009) reported that the nominal estimated uncertainty of total $NO_2$ columns is $0.27 \times 10^{15}$ molec.cm$^{-2}$ for the random part and $2.7 \times 10^{15}$ molec.cm$^{-2}$ for the systematic part, and an uncertainty of 20% is reported by comparisons with in-situ measurements (Verhoelst et al., 2021). However, the newer PGN $NO_2$ rnvs3p1-8 data, which are employed in this study, have considerably lower uncertainties due to changes in (1) the optical setup, (2) the gas-calibration approach and (3) a more accurate $NO_2$ effective temperature estimation. As reported in the PGN data products Readme (https://publications.pandonia-global-network.org/manuals/PGN_DataProducts_Readme.pdf), the combined uncertainty increases with decreasing SZA, reaching ~$0.45 \times 10^{15}$ molec.cm$^{-2}$ for $NO_2$ rnvs3p1-8 data and ~$1.2 \times 10^{15}$ molec.cm$^{-2}$ for HCHO rfus5p1-8 data at SZA=10° (median uncertainty over 137 data sets). The report uncertainty does not yet include the impact of spectral fitting quality and is therefore a lower limit. This uncertainty component will be included in a future PGN release; at Izana site, it is estimated to increase the reported uncertainty at SZA=10° to $1.0 \times 10^{15}$ molec.cm$^{-2}$ for $NO_2$ and $3.0 \times 10^{15}$ molec.cm$^{-2}$ for HCHO.
In this work, we use HCHO rfus5p1-8 and $NO_2$ rnvs3p1-8 direct sun total column measurements only from the ESA Validation Data Centre (EVDC) (https://evdc.esa.int, last access: 17 July 2024), because the PGN sub-dataset submitted to EVDC undergoes a more thorough quality check, in which the issues in PGN HCHO retrievals are mostly mitigated."
In Line 297-300, we added:
"For PGN data, we only use those with the flag "assured high quality" (data quality flag of 0) or "not-assured high quality" (data quality flag of 10) ((https://www.pandonia-global-network.org/wpcontent/uploads/2024/11/PGN_DataProducts_Readme_v1-8-9.pdf).”

Line 233: Grammatical suggestion in table caption, remove "the" before "alphabetical".

Thank you for the suggestion. We have removed it in the revised manuscript.

Line 249: How do the comparisons between satellite and MAX-DOAS/PGN change with the selection of time window and integration radius? It seems that all MAX-DOAS/PGN are in pollution hotspots. It would be interesting to see how these comparisons work in background areas particularly given the small influence of aerosols in those. Has this been explored during the comparison exercise?

Thank you very much for the comments. In principle, it is impossible to make "apple-to-apple" comparisons between satellite observations and ground-based measurements. To obtain conclusions that are as robust as possible, the collocation criteria, e.g., the average time window and spatial integration radius, should be carefully determined. Multiple factors need to be taken into consideration during the collocation processing, including the reduction of random uncertainties (especially for HCHO because of the low signal-to-noise ratio in the UV band), the diurnal variation characteristics and the spatial gradient of trace gases over the ground-based instruments.

Many previous studies have explored the effects of the collocation time window and integration radius. For example, Vigouroux et al. (2020) made comprehensive tests for HCHO validation. The effect of collocation time $\pm 6$ h to $\pm 3$ h (centered at the satellite overpass time) on the statistical bias is 4% at the Mexico City station, where the diurnal cycle amplitude is the greatest; Over many other stations, this effect is negligible due to the weak HCHO diurnal cycle. In terms of integration radius, 10 km is too small to give a sufficient number of coincidences and robust statistics, while the random spatial collocation error increases with collocation distance at all sites. For $NO_2$, the collocation criteria should be much stricter, considering the high signal-to-noise ratio of satellite observations in the Visible band, stronger $NO_2$ diurnal cycle, and stronger spatial gradient of $NO_2$ over polluted regions. Irie et al. (2008) showed differences of up to 25% in satellite $NO_2$ VCDs between pixels located 5 km to 50 km away from the site. Liu et al. (2020) also reported about 10% differences in the $NO_2$ validation statistics from the effect of time window and sampling distance.

Based on the experience from previous studies, we decided to select 11:00 – 16:00 LT ($\pm 2.5$ h) and 20 km as the collocation criteria for HCHO, and 13:00 – 14:00 LT ($\pm 0.5$ h) and 5 km for $NO_2$. Based on the selected time window, we tested the effect of integration radius on the statistics (slope, offset and correlation coefficient of the linear regression using the robust Theil-Sen estimator, as well as the normalized mean bias) using PGN measurements. As shown in Figure R1, the variation of the statistics with the sampling radius show consistent patterns between different satellite products and months. For HCHO (first row), as the sampling radius increases, the linear regression offset (second column) and negative normalized mean bias (NMB, fourth column) both increase, while the slope (first column) and the correlation coefficient (third column) reach maximum at 20 km. For $NO_2$ (second row), the sampling radius of 5 km is clearly the best choice for collocation, which presents the highest slope and correlation coefficient, as well as lowest offset and NMB.

For the second comment, most MAX-DOAS and PGN sites used in this study are in or near polluted regions, and only Altzomoni, Cape hedo, Davos and Fukue are remote sites regarding the $NO_2$ amount. For these four sites, the average time window and integration radius have little influence on the validation

results, because the NO$_2$ diurnal variation and spatial gradient are very weak over background areas. We also confirm that the influence of the aerosol correction is less significant there than at polluted sites with heavy aerosol loadings. By implementing the vertical smoothing process, we find that the agreement between satellite observations and ground-based measurements at Fukue and Cape hedo is greatly improved, indicating that a priori profiles are the dominant factor in the validation exercises over background areas.

[Figure]

**Figure R1.** Dependence of the slope (first column), offset (second column) and correlation coefficient (third column) of the linear regression using the robust Theil-Sen estimator, as well as the normalized mean bias (NMB, fourth column) on the sampling radius for HCHO and NO$_2$ validation based on PGN measurements. Red lines represent the results for July 2021, and blue lines for January 2022. Circles denote POMINO products and squares denote RPRO products.

Line 328: There are studies evaluating AMF and cloud pressure uncertainties. This is not the first time AMF uncertainties associated with cloud pressure are evaluated.

Thank you for the comment. We have added relevant references and removed the statement "which has never been discussed before" in the revised manuscript.

Line 380: It may be worth adding the work by Latsch et al., 2022 when discussing the characteristics of the different cloud products.

Thank you for the suggestion. We have added more sentences in the revised manuscript.
In Line 447-450, we added:
"The comparison results over China are also qualitatively consistent with the findings by Latsch et al. (2022), in which the ROCINN CRB cloud heights differ significantly from those of FRESCO-S when considering low cloud fraction and lowest cloud height values that are critical for tropospheric trace gas retrievals."

Section 4.1.2: This is an interesting analysis of the impact associated with different cloud top pressures. However, both products get cloud information from very different spectral ranges (440nm vs. 760nm)

and therefore they must be different. Given the wavelengths used in HCHO and NO$_2$ retrievals ROCINN cloud top pressure should be used with extreme caution. It would be good if the authors could comment on these considerations.

Thank you very much for the comment. We have added some sentences in the revised manuscript.
In Line 467-475, we added:
"In summary, the implementation of the cloud correction in HCHO and NO$_2$ retrievals is necessary, and the structural uncertainty due to different cloud parameters needs be taken into consideration in product comparisons. On the other hand, given the different spectral ranges used for trace gas retrievals (HCHO: 340 nm; NO$_2$: 440 nm) and cloud retrievals (OCRA/ROCINN-CRB: O$_2$-A band between 758 and 771 nm; FRESCO-S: O$_2$-A band around 760 nm), cloud parameters should always be used with caution, especially for low-cloud-fraction conditions. For example, in the ROCINN-CRB model, priori OCRA cloud fractions smaller than 0.05 are set to zero, and the ROCINN retrieval is not performed under such "clear-sky" conditions. The O$_2$-O$_2$ cloud algorithm uses the O$_2$-O$_2$ absorption window around 477 nm instead of the NIR spectral range, but it is more sensitive to low clouds and aerosols. Therefore, further work is still needed to address such discrepancies."

Section 4.3: Could the authors discuss the uncertainties associated with MODIS BRDF products and how the atmospheric correction in the BRDF retrieval accounts for aerosols or not? This could be relevant given the complicated nature of the BRDF atmospheric correction. What happens if the BRDF does not account for aerosols properly therefore has a bias that then is not considered in the POMINO aerosol correction.

Thank you very much for the comments. The uncertainty estimation and validation of MODIS BRDF/albedo products are difficult, given the complexity of the physical model and the scarcity of in-situ validation sites. According to the official statement, the accuracy of the high-quality MODIS operational BRDF/albedo products (MCD43) is less than 5% at most validation sites so far, and the albedo values with low quality flags have been found primarily within 10% of the field observations (https://modis-land.gsfc.nasa.gov/ValStatus.php?ProductID=MOD43). Besides, Chong et al. (2024) provide an estimate of random uncertainties in MODIS MCD43C1 surface reflectances for various surface types, which vary from 0.01 to 0.03 for most cases and reach more than 0.05 for surface with permanent snow and ice. We have added some sentences in the revised manuscript.
For the second comment, aerosol characteristics, including vertical profile, aerosol optical thickness (AOT), single scattering albedo (SSA), particle size distribution and refractive indices, are explicitly accounted for in the atmospheric correction procedure used for the MODIS surface reflectance retrieval. (Vermote et al., 2002; Vermote and Kotchenova, 2008). Therefore, there should not be a bias or overcorrection issue in the POMINO aerosol correction.
In Line 545-549, we added:
"The accuracy of the MODIS operational BRDF/albedo product (MCD43) is estimated by 5% to 10% of the filed data at most validation sites studied so far (https://modis-land.gsfc.nasa.gov/ValStatus.php?ProductID=MOD43). Chong et al. (2024) also provide an estimation of random uncertainties in MODIS MCD43C1 surface reflectances for various surface types, which vary in the range of 0.01 to 0.03 for most cases."

Section 4.4: Does the collocation of GEOS-CF, TM5 and MAX-DOAS observations follow the same methodology described above for TROPOMI data? Please clarify.

Yes, the collocation of trace gas vertical profiles between models and ground-based MAX-DOAS measurements follows the same methodology as described in Section 2.5. We have added a sentence to clarify it in the revised manuscript.
In Line 577-578, we added:
"The collocation of model profiles and MAX-DOAS profiles follows the same methodology as described in Sect. 2.5."

Section 5: The error analysis in Chong et al., 2024 (for BrO) and Ayazpour et al., 2024 (HCHO preprint) is done at an individual pixel level and includes information about the BRDF uncertainties relevant to this analysis. It may be worth commenting on their results.

Thank you for the suggestion. We have added comments in the revised manuscript.
In Line 545-549, we added:
"The accuracy of the MODIS operational BRDF/albedo product (MCD43) is estimated by 5% to 10% of the filed data at most validation sites studied so far (https://modis-land.gsfc.nasa.gov/ValStatus.php?ProductID=MOD43). Chong et al. (2024) also provide an estimation of random uncertainties in MODIS MCD43C1 surface reflectances for various surface types, which vary in the range of 0.01 to 0.03 for most cases."
In Line 680-683, we added:
"Quantification of the errors at an individual pixel level have been achieved in previous studies (Boersma et al., 2004; Chong et al., 2024; Van Geffen et al., 2022b). As an alternative option to the Gaussian error propagation method, artificial-intelligence-based methods are an appealing approach to be tried in our future work."

Table S1: The reference spectrum $I_0$ in the case of $NO_2$ retrievals is a solar irradiance recorded by TROPOMI (van Geffen et al., 2020)

Thank you very much for the comment. We have corrected the description in the revised Supplement.
For the reference spectrum for $NO_2$ in Table S1, we have corrected it to be "Daily measured solar spectrum from TROPOMI"
Besides, we have also added another entry named "High-resolution solar irradiance spectrum for wavelength calibration" in Table S1, and it is a high-resolution solar reference spectrum from Chance and Kurucz (2010) for both HCHO and $NO_2$.

Figure S1: MODIS MCD43C2.061 product has a resolution of 0.05 x 0.05.

Thank you for the comment. In the previous version of POMINO-TROPOMI algorithm, of which the retrieval domain is limited to Asia, the spatial resolution of MODIS MCD43C2.061 BRDF coefficients for retrieval is $0.05° \times 0.05°$. In this work, because we extended the retrieval domain to the whole globe, we decided to low the spatial resolution to $0.25° \times 0.25°$ to reduce the data storage.

**Authors' response to comments from Anonymous Referee #3**

**General comments:**

This paper presents global tropospheric HCHO and $NO_2$ vertical column retrievals from TROPOMI, utilizing the POMINO algorithm developed by Peking University, which focuses on improvements in air mass factor (AMF) calculations. The study includes comprehensive sensitivity tests on AMF input parameters. The research topic fits well in the scope of AMT and is well-structured, providing valuable scientific insights.

We sincerely thank the Referee #3 for reviewing our paper and providing constructive comments for improvement. Responses to these comments are provided below.

**Specific comments:**

1. While I understand the reason for using the TROPOMI RPRO product for the study period, "general readers" may not be familiar with the differences between the OFFL and RPRO v2.4.1 products (e.g., the application of the reprocessed Level 1 version) and the specific improvements reflected in RPRO products. Including a brief explanation of RPRO v2.4.1 and its distinctions from the OFFL product would enhance the clarity of the manuscript.

Thank you for the suggestion. We have added more explanations in Section 2.2 in the revised manuscript. In Line 178-185, we added:
"Compared to the previous HCHO v2.3.0 processor, HCHO v2.4.1 processor uses new improved Level 1b v2.1.0 data products as input, and has been applied for a full mission reprocessing starting from 7th May 2018. For $NO_2$, the improvements of the v2.4.0 processor include the use of a DLER climatology derived from TROPOMI observations and new improved Level 1b v2.1.0 data products as input, which has also been used for a full mission reprocessing from 1st May 2018. Detailed information of S5P TROPOMI L2 HCHO and $NO_2$ processing baseline, including the processor version, in operation period and relevant improvements can be found at https://sentiwiki.copernicus.eu/web/s5p-processing."

2. Although the flow chart of global POMINO-TROPOMI algorithm is included in the supplementary material (Fig. S1), it would be helpful to mention in a sentence in Section 2.1 or 2.2 that the POMINO-TROPOMI algorithm performs only sensitivity tests on AMF, using tropospheric slant columns directly from the RPRO products. It should be clarified that slant column retrieval and stratosphere-troposphere separation are not part of the POMINO-TROPOMI algorithm in this study.

We have added a sentence to clarify it in the revised manuscript.
In Line 176-178, we added:
"The DOAS spectral fit, HCHO dSCD background correction and $NO_2$ stratosphere-troposphere separation are not included in this study, so corrected HCHO dSCDs and tropospheric $NO_2$ SCDs are directly taken from the RPRO HCHO v2.4.1 product and RPRO $NO_2$ v2.4.0 product, respectively."

3. In this study, you used cloud top pressure from FRESCO-S and recalculated the cloud fraction at 340 nm (for HCHO) and 440 nm (for NO$_2$) by simulating the TOA reflectance using auxiliary input parameters. As FRESCO cloud products does not explicitly correct for the presence of aerosols but retrieve parameters based on the O$_2$-O$_2$ absorption, aerosols are implicitly included in the FRESCO. This raises a concern that the POMINO-TROPOMI algorithm may "overcorrect" for aerosols here. I recommend providing a more detailed explanation of the potential overcorrection effects on aerosols and their impact on the derived results.

Thank you very much for this comment, and a similar comment was raised by Referee #1. Note that the FRESCO-S does not retrieve cloud parameters based on the O$_2$-O$_2$ absorption, but based on the O$_2$ A-band around 760 nm.

We agree that the aerosol overcorrection issue occurs for partly cloudy pixels because we only recalculate the cloud fraction with explicit aerosol corrections, but use the cloud top pressure from FRESCO-S product which already implicitly includes aerosols. Liu et al. (2020) quantified such overcorrection issue for aerosols by conducting a sensitivity study using a "semi-explicit" aerosol correction approach. In this approach, aerosol optical effects are explicitly corrected for clear-sky AMFs, but are excluded for the cloudy-sky portion of partly cloudy pixels. Results show that NO$_2$ differences due to the aerosol correction choice for cloudy-sky AMFs vary from 3.1% to 11.2% over East Asia in July 2018, depending on the NO$_2$ pollution level (Section 3.3 in Liu et al., 2020). It should be noted that the FRESCO-S cloud top pressure data stored in the v1.2–v1.3 NO$_2$ data product, as used by Liu et al. (2020), are reported with a high bias over scenes with low cloud fractions and/or a considerable aerosol load. An improved version based on the FRESCO-wide approach is applied in v1.4 and subsequent NO$_2$ products, which was proven to be more realistic compared with the old version (Van Geffen et al., 2022a, b).

Using the updated FRESCO-S cloud top pressure data stored in the RPRO v2.4.0 NO$_2$ product, we attempt to estimate the impact of the aerosol overcorrection issue without conducting a sensitivity study. Given the fact that, in the retrieval algorithm, the cloud is assumed to be an optically thick Lambertian reflector with a high albedo of 0.8, it is reasonable to assume that the impact of this overcorrection becomes non-negligible when the FRESCO-S cloud top pressure is too high, meaning the cloud is very close to the surface and therefore vertically mixed with aerosols and trace gases; while for pixels where the cloud is higher than the trace gases, the aerosol correction should have little influence on the cloudy-sky AMFs. Based on this strategy, we selected all valid pixels where the difference between the surface pressure and the FRESCO-S cloud top pressure is equal to 100 hPa or less in July 2021 and January 2022. In such case, we can mitigate the potential aerosol overcorrection by using aerosol-corrected clear-sky AMFs instead of aerosol-corrected total AMFs.

The comparison result in Figure S6 (shown below) shows that the normalized mean bias (NMB) is around 14% on average for HCHO retrievals (~16% for clean pixels with HCHO column $\leq 10 \times 10^{15}$ molec.cm$^{-2}$, and ~8% for polluted pixels with HCHO column $> 10 \times 10^{15}$ molec.cm$^{-2}$), and around 8% on average for NO$_2$ retrievals. The NO$_2$ results are also qualitatively in line with those in Liu et al. (2020). Therefore, we tentatively estimate the uncertainty due to the aerosol overcorrection to be in the range from 10% to 15% for HCHO and 10% for NO$_2$.

In line 403-429 in the revised manuscript, we added:

"One issue existing in the process of cloud correction in the POMINO retrieval is that only the cloud fraction is re-calculated with explicit aerosol corrections, while the cloud top pressure is taken from the external dataset, i.e., the FRESCO-S cloud product, in which the aerosols are implicitly accounted for.

As a result, this step introduces presumably an aerosol overcorrection issue in the cloud top pressures of partly cloudy pixels, and therefore brings in additional uncertainties in the AMF calculations. Lin et al. (2015) reported that excluding aerosols leads to an increase of $O_2$-$O_2$-based cloud top pressures (from 700–900 hPa to 750–950 hPa) over eastern China, but it is difficult to clarify the mechanism due to its complexity (Lin et al., 2014). Currently there is no direct way to estimate the effect of aerosol correction on the FRESCO-S cloud height retrieval without doing $O_2$ A-band cloud retrieval tests, which is beyond the scope of this study. However, below we give an estimation of the uncertainty in POMINO HCHO and $NO_2$ vertical columns caused by this issue.

Given the fact that, in the retrieval algorithm, the cloud is assumed to be an optically thick Lambertian reflector with a high albedo of 0.8, the cloudy-sky AMF (and hence tropospheric AMF) is very sensitive to the accuracy of the cloud height when the cloud is low and vertically mixed with the aerosols and trace gases. In these cases, we can assume that the retrieved cloud height is primarily influenced by aerosols (Van Geffen et al., 2022a), therefore the aerosol overcorrection issue becomes non-negligible. Focusing on valid pixels for which the difference between the surface pressure and the FRESCO-S cloud top pressure is equal to 100 hPa or less (~17.5% and ~19.9% of total pixels in July 2021 and January 2022, respectively), the aerosol overcorrection uncertainty can be roughly estimated from the difference of HCHO and $NO_2$ vertical columns retrieved using either aerosol-corrected clear-sky AMFs (aerosol correction applied; cloud correction not applied) or aerosol-corrected total AMFs (both aerosol and cloud corrections applied). Based on the results shown in Figure S6, we tentatively estimate the uncertainty to be in the range from 10% to 15% for HCHO, and within 10% for $NO_2$. The estimated $NO_2$ uncertainty level is also supported by the sensitivity test results in Liu et al. (2020). They implemented a "semi-explicit" aerosol correction approach, in which aerosol optical effects are explicitly corrected for clear-sky AMFs, but are excluded for the cloudy-sky portion of partly cloudy pixels, and found the $NO_2$ differences due to the aerosol correction choice for cloudy-sky AMFs vary from 3.1% to 11.2% over eastern China in July 2018. The tentatively estimated uncertainty range above is comparable to or less than that from other ancillary parameters (Sect. 5), and only needs to be taken into account for partly cloudy pixels with low clouds."

In the revised Supplement, we added:

[Figure]

**Figure S6.** Scatterplots of POMINO tropospheric HCHO (**a** and **b**) and NO$_2$ (**c** and **d**) VCDs retrieved using either aerosol-corrected and cloud-corrected total AMF (x-axis) or aerosol-corrected clear-sky AMF (y-axis), from all pixels where the difference between surface pressure and FRESCO-S cloud top pressure is equal to 100 hPa or less. The left column shows the results for July 2021, and the right column for January 2022. The slope, offset and correlation from a linear regression using the robust Theil-Sen estimator and normalized mean bias (NMB) are given in each panel and plotted as the red line. The black dashed line is the 1:1 line.

4. In Sect. 4.1.1, for both HCHO and NO$_2$, the differences of clear-sky AMF and total AMF across cloud top pressure ranges (Fig. 3) using all global pixels in both summer (July 2021) and winter (January 2022) shows different patterns (negative and positive) based on the 700 hPa cloud top pressure threshold. These patterns include the combined effects of seasonal and global variations. Could you describe if the cloud correction pattern for cloud top pressure differs depending on season (e.g., summer vs winter) and region (e.g., polluted vs clean)?

Thank you very much for the comment. Figure R1 specifies the cloud correction pattern for clean and polluted HCHO pixels in July 2021 and January 2022, based on the FRESCO-S cloud top pressures and POMINO re-calculated cloud fractions at 440 nm with explicit aerosol corrections. Figure R2 is similar to Fig. R1 but for NO$_2$. The pixel is determined over a polluted region if its tropospheric HCHO or NO$_2$ column exceeds the 50 percentiles of all the observations in that month.

[Figure]

**Figure R1.** Differences of HCHO clear-sky AMF to total AMF for different cloud radiance fraction with an interval of 0.05 in different cloud top pressure ranges (shown in different colors). All pixels with HCHO QA > 0.5 are included. The first row is the result in July 2021, and the second row in January 2022; The left column is the result for clean pixels, and the right column for polluted pixels.

In general, for both HCHO and $NO_2$, the cloud correction patterns are similar and nearly independent on the season or the pollution level. This is expected because the occurrence frequency of each dot in Figs. R1 and R2 are dependent on the season and the pollution level, but the differences between clear-sky AMF and total AMF are only determined by the relative height of the cloud to the trace gas molecules. For example, the occurrence frequency of high clouds should be higher in summertime than in wintertime due to stronger vertical convections, but the effect of the cloud correction could be similar if the relative height of the cloud to the trace gases is similar for pixels in different situations. Therefore, we decided to show the results for global valid pixels in both months for the sake of brevity.

[Figure]

**Figure R2.** Differences of $NO_2$ clear-sky AMF to total AMF for different cloud radiance fraction with an interval of 0.05 in different cloud top pressure ranges (shown in different colors). All pixels with $NO_2$ QA > 0.5 are included. The first row is the result in July 2021, and the second row in January 2022; The left column is the result for clean pixels, and the right column for polluted pixels.

5. In Sect. 4.4, the descriptions of the chemistry transport models GEOS-CF and TM5-MP are insufficient. Please provide more detailed information on the specifications of the CTMs, such as vertical resolution, tropospheric chemistry, emissions, and meteorological fields and so on. Additionally, please include relevant references.

Thank you for the comment. We have added a detailed comparison of GEOS-CF and TM5-MP in the revised manuscript and supplement.

In Line 199-200 in the revised manuscript, we added:

"Detailed comparison of the specifications between GEOS-CF and TM5-MP is provided in Table S2."

In the revised Supplement, we added:

**Table S2.** Comparison of the specifications between GEOS-CF and TM5-MP.

| Specification | GEOS-CF (Keller et al., 2021) | TM5-MP (Huijnen et al., 2010; Williams et al., 2017) |
|---|---|---|
| Resolution | Horizontal: 0.25° × 0.25°
 Vertical: 72 hybrid-eta levels | Horizonal: 1° × 1°
 Vertical: 34 layers |
| Meteorological field | GEOS-FP for instrument teams (GEOS FP-IT; https://gmao.gsfc.nasa.gov/pubs/docs/Lucchesi865.pdf) | ERA-Interim re-analysis (Dee et al., 2011) |
| $NO_x$ & VOC emissions | (1) Anthropogenic: HTAP v2.2 (Janssens-Maenhout et al., 2015); RETRO (Schultz et al., 2008); DICE-Africa (Marais and Wiedinmyer, 2016)
 (2) Aircraft: AEIC Stettler et al. (2011)
 (3) Biomass burning: QFED v2.5 (https://ntrs.nasa.gov/citations/20180005253)
 (4) Lightning $NO_x$: Murray et al. (2012)
 (5) Soil $NO_x$: Hudman et al. (2012)
 (6) Biogenic VOCs: MEGAN v2.1 (Guenther et al., 2012) | (1) Anthropogenic: MACCity (Granier et al., 2011)
 (2) Aircraft (only for NO): a homogeneous hourly flux estimate
 (3) Biomass burning: GFED v3 (van der Werf et al., 2010)
 (4) Lightning $NO_x$: parameterization using convective precipitation fields (Meijer et al., 2001) with the constraint on the annual global emission term at ~ 6Tg N $yr^{-1}$
 (5) Biogenic component: CLM-MEGAN v2.1 (Zeng et al., 2015); MEGAN (Sindelarova et al., 2014) |
| Chemistry | (1) Full tropospheric chemistry for NOx + HOx + VOC + $O_3$ + halogen + aerosols (https://wiki.seas.harvard.edu/geos-chem/index.php?title=Simulations_using_KPP-built_mechanisms)
 (2) Stratospheric chemistry fully coupled with the tropospheric chemistry through the Unified tropospheric-stratospheric Chemistry eXtension (UCX; Eastham et al., 2014) | (1) Modified CB05 (mCB05) chemical mechanism for gas-phase chemistry (Williams et al., 2013)
 (2) No aerosol scheme
 (3) no explicit stratospheric chemistry |
| Advection scheme | Finite-volume dynamical core (Lin, 2004) with a cubed sphere grid discretization (Putman and Lin, 2007) | Slopes scheme (Russell and Lerner, 1981) |

| | | |
|---|---|---|
| Convection Scheme | Relaxed Arakawa-Schubert scheme (Moorthi and Suarez, 1992) | Convective mass fluxes and detrainment rates from the ERA-Interim re-analysis (Dee et al., 2011) |
| Boundary layer diffusion | Non-local Lock scheme (Lock et al., 2000) interfaced with the Richardson-number-based scheme of Louis and Geleyn (J-F. Louis et al., 1982) | Holtslag and Boville (1993) |

6. In Sect. 5, please provide a summary table for estimates of the contributions to the AMF uncertainties from individual error sources for HCHO and NO$_2$ retrievals.

We have added the summary tables for the estimated uncertainty budget of POMINO HCHO and NO$_2$ vertical columns in the revised manuscript.

In Line 662-664, we added:

**Table 3.** Estimated uncertainty budget of POMINO HCHO vertical columns for monthly mean low and elevated columns (higher than $10 \times 10^{15}$ molec.cm$^{-2}$).

| | Remote regions / low columns | Elevated column regions / periods |
|---|---|---|
| Differential slant column uncertainties (De Smedt, 2022) | 25% | 15% |
| Background correction uncertainties (De Smedt, 2022) | 40% | 10% |
| ● dSCD normalization uncertainties | $0 - 4 \times 10^{15}$ molec.cm$^{-2}$ | |
| ● model background uncertainties | $0 - 2 \times 10^{15}$ molec.cm$^{-2}$ | |
| AMF uncertainties | 70% | 30% |
| ● from a priori profiles uncertainties | 60% | 20% |
| ● from aerosol correction uncertainties | 5% | 10% |
| ● from surface reflectance uncertainties | 20% | 10% |
| ● from cloud correction uncertainties | 20% | 10% |
| ● from aerosol overcorrection issue uncertainties (only for partly cloudy pixels with low clouds) | 15% | 10% |
| Tropospheric vertical column uncertainty | 85% | 35% |

In Line 673-674, we added:

**Table 4.** Estimated uncertainty budget of monthly mean POMINO NO$_2$ vertical columns.

| | All regions |
|---|---|
| Total slant column uncertainties (Van Geffen et al., 2022b) | $0.5 - 0.6 \times 10^{15}$ molec.cm$^{-2}$ |
| Stratospheric slant column uncertainties (Van Geffen et al., 2022b) | $0.2 \times 10^{15}$ molec.cm$^{-2}$ |
| AMF uncertainties | 25% – 30% |
| ● from a priori profiles uncertainties | 10% |
| ● from aerosol correction uncertainties | 10% – 20% |
| ● from surface reflectance uncertainties | 10% |
| ● from cloud correction uncertainties | 10% |
| ● from aerosol overcorrection issue uncertainties | 10% |

| | |
|---|---|
| (only for partly cloudy pixels with low clouds) | |
| Tropospheric vertical column uncertainty | $0.3 \times 10^{15}$ molec.cm$^{-2}$ + [0.2 to 0.4] × VCD |

Note: the uncertainty in the total slant columns is mostly absorbed by the stratosphere-troposphere separation step, and may not propagate into the tropospheric slant columns. (Van Geffen et al., 2015)

Table 1: there is a typo in POMINO HCHO CF: re-calculated at 340 nm.

Regarding the cloud fraction in POMINO HCHO retrievals, we use values re-calculated at 440 nm (in the NO$_2$ retrieval) instead of re-calculating them at 340 nm, because (1) the cloud fraction derived at 440 nm is expected to be more reliable than at 340 nm due to the larger noise in the UV band; (2) we want to perform fully consistent cloud corrections in both POMINO HCHO and NO$_2$ retrievals. We have changed the description to "CF and CP: same as POMINO NO$_2$" in Table 1 to make it more clear in the revised manuscript.

Figure 3: Please add the unit for cloud top pressure [hPa]

Done.

**References:**

Chong, H., González Abad, G., Nowlan, C. R., Chan Miller, C., Saiz-Lopez, A., Fernandez, R. P., Kwon, H.-A., Ayazpour, Z., Wang, H., Souri, A. H., Liu, X., Chance, K., O'Sullivan, E., Kim, J., Koo, J.-H., Simpson, W. R., Hendrick, F., Querel, R., Jaross, G., Seftor, C., and Suleiman, R. M.: Global retrieval of stratospheric and tropospheric BrO columns from the Ozone Mapping and Profiler Suite Nadir Mapper (OMPS-NM) on board the Suomi-NPP satellite, Atmospheric Measurement Techniques, 17, 2873–2916, https://doi.org/10.5194/amt-17-2873-2024, 2024.

De Smedt, I.: TROPOMI ATBD of HCHO data products version 2.4.1, 2022.

De Smedt, I., Pinardi, G., Vigouroux, C., Compernolle, S., Bais, A., Benavent, N., Boersma, F., Chan, K.-L., Donner, S., Eichmann, K.-U., Hedelt, P., Hendrick, F., Irie, H., Kumar, V., Lambert, J.-C., Langerock, B., Lerot, C., Liu, C., Loyola, D., Piters, A., Richter, A., Rivera Cárdenas, C., Romahn, F., Ryan, R. G., Sinha, V., Theys, N., Vlietinck, J., Wagner, T., Wang, T., Yu, H., and Van Roozendael, M.: Comparative assessment of TROPOMI and OMI formaldehyde observations and validation against MAX-DOAS network column measurements, Atmospheric Chemistry and Physics, 21, 12561–12593, https://doi.org/10.5194/acp-21-12561-2021, 2021.

Dee, D. P., Uppala, S. M., Simmons, A. J., Berrisford, P., Poli, P., Kobayashi, S., Andrae, U., Balmaseda, M. A., Balsamo, G., Bauer, P., Bechtold, P., Beljaars, A. C. M., van de Berg, L., Bidlot, J., Bormann, N., Delsol, C., Dragani, R., Fuentes, M., Geer, A. J., Haimberger, L., Healy, S. B., Hersbach, H., Hólm, E. V., Isaksen, L., Kållberg, P., Köhler, M., Matricardi, M., McNally, A. P., Monge-Sanz, B. M., Morcrette, J.-J., Park, B.-K., Peubey, C., de Rosnay, P., Tavolato, C., Thépaut, J.-N., and Vitart, F.: The ERA-Interim reanalysis: configuration and performance of the data assimilation system, Quarterly Journal of the Royal Meteorological Society, 137, 553–597, https://doi.org/10.1002/qj.828, 2011.

Eastham, S. D., Weisenstein, D. K., and Barrett, S. R. H.: Development and evaluation of the unified tropospheric–stratospheric chemistry extension (UCX) for the global chemistry-transport model

GEOS-Chem, Atmospheric Environment, 89, 52–63, https://doi.org/10.1016/j.atmosenv.2014.02.001, 2014.

Granier, C., Bessagnet, B., Bond, T., D'Angiola, A., Denier van der Gon, H., Frost, G. J., Heil, A., Kaiser, J. W., Kinne, S., Klimont, Z., Kloster, S., Lamarque, J.-F., Liousse, C., Masui, T., Meleux, F., Mieville, A., Ohara, T., Raut, J.-C., Riahi, K., Schultz, M. G., Smith, S. J., Thompson, A., van Aardenne, J., van der Werf, G. R., and van Vuuren, D. P.: Evolution of anthropogenic and biomass burning emissions of air pollutants at global and regional scales during the 1980–2010 period, Climatic Change, 109, 163, https://doi.org/10.1007/s10584-011-0154-1, 2011.

Guenther, A. B., Jiang, X., Heald, C. L., Sakulyanontvittaya, T., Duhl, T., Emmons, L. K., and Wang, X.: The Model of Emissions of Gases and Aerosols from Nature version 2.1 (MEGAN2.1): an extended and updated framework for modeling biogenic emissions, Geoscientific Model Development, 5, 1471–1492, https://doi.org/10.5194/gmd-5-1471-2012, 2012.

Herman, J., Cede, A., Spinei, E., Mount, G., Tzortziou, M., and Abuhassan, N.: NO2 column amounts from ground-based Pandora and MFDOAS spectrometers using the direct-sun DOAS technique: Intercomparisons and application to OMI validation, Journal of Geophysical Research: Atmospheres, 114, https://doi.org/10.1029/2009JD011848, 2009.

Holtslag, A. a. M. and Boville, B. A.: Local Versus Nonlocal Boundary-Layer Diffusion in a Global Climate Model, 1993.

Hudman, R. C., Moore, N. E., Mebust, A. K., Martin, R. V., Russell, A. R., Valin, L. C., and Cohen, R. C.: Steps towards a mechanistic model of global soil nitric oxide emissions: implementation and space based-constraints, Atmospheric Chemistry and Physics, 12, 7779–7795, https://doi.org/10.5194/acp-12-7779-2012, 2012.

Huijnen, V., Williams, J., van Weele, M., van Noije, T., Krol, M., Dentener, F., Segers, A., Houweling, S., Peters, W., de Laat, J., Boersma, F., Bergamaschi, P., van Velthoven, P., Le Sager, P., Eskes, H., Alkemade, F., Scheele, R., Nédélec, P., and Pätz, H.-W.: The global chemistry transport model TM5: description and evaluation of the tropospheric chemistry version 3.0, Geoscientific Model Development, 3, 445–473, https://doi.org/10.5194/gmd-3-445-2010, 2010.

Irie, H., Kanaya, Y., Akimoto, H., Tanimoto, H., Wang, Z., Gleason, J. F., and Bucsela, E. J.: Validation of OMI tropospheric $NO_2$ column data using MAX-DOAS measurements deep inside the North China Plain in June 2006: Mount Tai Experiment 2006, Atmospheric Chemistry and Physics, 8, 6577–6586, https://doi.org/10.5194/acp-8-6577-2008, 2008.

Janssens-Maenhout, G., Crippa, M., Guizzardi, D., Dentener, F., Muntean, M., Pouliot, G., Keating, T., Zhang, Q., Kurokawa, J., Wankmüller, R., Denier van der Gon, H., Kuenen, J. J. P., Klimont, Z., Frost, G., Darras, S., Koffi, B., and Li, M.: HTAP_v2.2: a mosaic of regional and global emission grid maps for 2008 and 2010 to study hemispheric transport of air pollution, Atmospheric Chemistry and Physics, 15, 11411–11432, https://doi.org/10.5194/acp-15-11411-2015, 2015.

J-F. Louis, M. Tiedtke, and J.-F. Geleyn: A short history of the PBL parameterization at ECMWF, in: Workshop on Planetary Boundary Layer parameterization, 25-27 November 1981, Workshop on Planetary Boundary Layer parameterization, 25-27 November 1981, Shinfield Park, Reading, 59–79, 1982.

Keller, C. A., Knowland, K. E., Duncan, B. N., Liu, J., Anderson, D. C., Das, S., Lucchesi, R. A., Lundgren, E. W., Nicely, J. M., Nielsen, E., Ott, L. E., Saunders, E., Strode, S. A., Wales, P. A., Jacob, D. J., and Pawson, S.: Description of the NASA GEOS Composition Forecast Modeling System GEOS-CF v1.0, Journal of Advances in Modeling Earth Systems, 13, e2020MS002413,

https://doi.org/10.1029/2020MS002413, 2021.

Lin, S.-J.: A "Vertically Lagrangian" Finite-Volume Dynamical Core for Global Models, 2004.

Liu, M., Lin, J., Kong, H., Boersma, K. F., Eskes, H., Kanaya, Y., He, Q., Tian, X., Qin, K., Xie, P., Spurr, R., Ni, R., Yan, Y., Weng, H., and Wang, J.: A new TROPOMI product for tropospheric $NO_2$ columns over East Asia with explicit aerosol corrections, Atmospheric Measurement Techniques, 13, 4247–4259, https://doi.org/10.5194/amt-13-4247-2020, 2020.

Lock, A. P., Brown, A. R., Bush, M. R., Martin, G. M., and Smith, R. N. B.: A New Boundary Layer Mixing Scheme. Part I: Scheme Description and Single-Column Model Tests, 2000.

Marais, E. A. and Wiedinmyer, C.: Air Quality Impact of Diffuse and Inefficient Combustion Emissions in Africa (DICE-Africa), Environ. Sci. Technol., 50, 10739–10745, https://doi.org/10.1021/acs.est.6b02602, 2016.

Meijer, E. W., van Velthoven, P. F. J., Brunner, D. W., Huntrieser, H., and Kelder, H.: Improvement and evaluation of the parameterisation of nitrogen oxide production by lightning, Physics and Chemistry of the Earth, Part C: Solar, Terrestrial & Planetary Science, 26, 577–583, https://doi.org/10.1016/S1464-1917(01)00050-2, 2001.

Moorthi, S. and Suarez, M. J.: Relaxed Arakawa-Schubert. A Parameterization of Moist Convection for General Circulation Models, 1992.

Murray, L. T., Jacob, D. J., Logan, J. A., Hudman, R. C., and Koshak, W. J.: Optimized regional and interannual variability of lightning in a global chemical transport model constrained by LIS/OTD satellite data, Journal of Geophysical Research: Atmospheres, 117, https://doi.org/10.1029/2012JD017934, 2012.

Putman, W. M. and Lin, S.-J.: Finite-volume transport on various cubed-sphere grids, Journal of Computational Physics, 227, 55–78, https://doi.org/10.1016/j.jcp.2007.07.022, 2007.

Russell, G. L. and Lerner, J. A.: A New Finite-Differencing Scheme for the Tracer Transport Equation, 1981.

Schultz, M. G., Heil, A., Hoelzemann, J. J., Spessa, A., Thonicke, K., Goldammer, J. G., Held, A. C., Pereira, J. M. C., and van het Bolscher, M.: Global wildland fire emissions from 1960 to 2000, Global Biogeochemical Cycles, 22, https://doi.org/10.1029/2007GB003031, 2008.

Sindelarova, K., Granier, C., Bouarar, I., Guenther, A., Tilmes, S., Stavrakou, T., Müller, J.-F., Kuhn, U., Stefani, P., and Knorr, W.: Global data set of biogenic VOC emissions calculated by the MEGAN model over the last 30 years, Atmospheric Chemistry and Physics, 14, 9317–9341, https://doi.org/10.5194/acp-14-9317-2014, 2014.

Stettler, M. E. J., Eastham, S., and Barrett, S. R. H.: Air quality and public health impacts of UK airports. Part I: Emissions, Atmospheric Environment, 45, 5415–5424, https://doi.org/10.1016/j.atmosenv.2011.07.012, 2011.

Van Geffen, J., Eskes, H., Compernolle, S., Pinardi, G., Verhoelst, T., Lambert, J.-C., Sneep, M., ter Linden, M., Ludewig, A., Boersma, K. F., and Veefkind, J. P.: Sentinel-5P TROPOMI $NO_2$ retrieval: impact of version v2.2 improvements and comparisons with OMI and ground-based data, Atmospheric Measurement Techniques, 15, 2037–2060, https://doi.org/10.5194/amt-15-2037-2022, 2022a.

Van Geffen, J. H. G. M., Boersma, K. F., Van Roozendael, M., Hendrick, F., Mahieu, E., De Smedt, I., Sneep, M., and Veefkind, J. P.: Improved spectral fitting of nitrogen dioxide from OMI in the 405–465 nm window, Atmospheric Measurement Techniques, 8, 1685–1699, https://doi.org/10.5194/amt-8-1685-2015, 2015.

Van Geffen, J. H. G. M., Eskes, H. J., Boersma, K. F., and Veefkind, P.: TROPOMI ATBD of the total

and tropospheric NO2 data products version 2.4.0, 2022b.

Verhoelst, T., Compernolle, S., Pinardi, G., Lambert, J.-C., Eskes, H. J., Eichmann, K.-U., Fjæraa, A. M., Granville, J., Niemeijer, S., Cede, A., Tiefengraber, M., Hendrick, F., Pazmiño, A., Bais, A., Bazureau, A., Boersma, K. F., Bognar, K., Dehn, A., Donner, S., Elokhov, A., Gebetsberger, M., Goutail, F., Grutter de la Mora, M., Gruzdev, A., Gratsea, M., Hansen, G. H., Irie, H., Jepsen, N., Kanaya, Y., Karagkiozidis, D., Kivi, R., Kreher, K., Levelt, P. F., Liu, C., Müller, M., Navarro Comas, M., Piters, A. J. M., Pommereau, J.-P., Portafaix, T., Prados-Roman, C., Puentedura, O., Querel, R., Remmers, J., Richter, A., Rimmer, J., Rivera Cárdenas, C., Saavedra de Miguel, L., Sinyakov, V. P., Stremme, W., Strong, K., Van Roozendael, M., Veefkind, J. P., Wagner, T., Wittrock, F., Yela González, M., and Zehner, C.: Ground-based validation of the Copernicus Sentinel-5P TROPOMI NO$_2$ measurements with the NDACC ZSL-DOAS, MAX-DOAS and Pandonia global networks, Atmospheric Measurement Techniques, 14, 481–510, https://doi.org/10.5194/amt-14-481-2021, 2021.

Vermote, E. F. and Kotchenova, S.: Atmospheric correction for the monitoring of land surfaces, Journal of Geophysical Research: Atmospheres, 113, https://doi.org/10.1029/2007JD009662, 2008.

Vermote, E. F., El Saleous, N. Z., and Justice, C. O.: Atmospheric correction of MODIS data in the visible to middle infrared: first results, Remote Sensing of Environment, 83, 97–111, https://doi.org/10.1016/S0034-4257(02)00089-5, 2002.

Vigouroux, C., Langerock, B., Bauer Aquino, C. A., Blumenstock, T., Cheng, Z., De Mazière, M., De Smedt, I., Grutter, M., Hannigan, J. W., Jones, N., Kivi, R., Loyola, D., Lutsch, E., Mahieu, E., Makarova, M., Metzger, J.-M., Morino, I., Murata, I., Nagahama, T., Notholt, J., Ortega, I., Palm, M., Pinardi, G., Röhling, A., Smale, D., Stremme, W., Strong, K., Sussmann, R., Té, Y., van Roozendael, M., Wang, P., and Winkler, H.: TROPOMI–Sentinel-5 Precursor formaldehyde validation using an extensive network of ground-based Fourier-transform infrared stations, Atmospheric Measurement Techniques, 13, 3751–3767, https://doi.org/10.5194/amt-13-3751-2020, 2020.

van der Werf, G. R., Randerson, J. T., Giglio, L., Collatz, G. J., Mu, M., Kasibhatla, P. S., Morton, D. C., DeFries, R. S., Jin, Y., and van Leeuwen, T. T.: Global fire emissions and the contribution of deforestation, savanna, forest, agricultural, and peat fires (1997–2009), Atmospheric Chemistry and Physics, 10, 11707–11735, https://doi.org/10.5194/acp-10-11707-2010, 2010.

Williams, J. E., van Velthoven, P. F. J., and Brenninkmeijer, C. a. M.: Quantifying the uncertainty in simulating global tropospheric composition due to the variability in global emission estimates of Biogenic Volatile Organic Compounds, Atmospheric Chemistry and Physics, 13, 2857–2891, https://doi.org/10.5194/acp-13-2857-2013, 2013.

Williams, J. E., Boersma, K. F., Le Sager, P., and Verstraeten, W. W.: The high-resolution version of TM5-MP for optimized satellite retrievals: description and validation, Geoscientific Model Development, 10, 721–750, https://doi.org/10.5194/gmd-10-721-2017, 2017.

Zeng, G., Williams, J. E., Fisher, J. A., Emmons, L. K., Jones, N. B., Morgenstern, O., Robinson, J., Smale, D., Paton-Walsh, C., and Griffith, D. W. T.: Multi-model simulation of CO and HCHO in the Southern Hemisphere: comparison with observations and impact of biogenic emissions, Atmospheric Chemistry and Physics, 15, 7217–7245, https://doi.org/10.5194/acp-15-7217-2015, 2015.